# PTP4A1 promotes TGFβ signaling and fibrosis in systemic sclerosis

Cristiano Sacchetti[1,2], Yunpeng Bai[3], Stephanie M. Stanford[1,2], Paola Di Benedetto[4], Paola Cipriani[4], Eugenio Santelli[1,2], Sonsoles Piera-Velazquez[5], Vladimir Chernitskiy[6], William B. Kiosses[7], Arnold Ceponis[1], Klaus H. Kaestner[8], Francesco Boin[6], Sergio A. Jimenez[5], Roberto Giacomelli[4], Zhong-Yin Zhang[3] & Nunzio Bottini[1,2]

Systemic sclerosis (SSc) is an autoimmune disease characterized by fibrosis of skin and internal organs. Protein tyrosine phosphatases have received little attention in the study of SSc or fibrosis. Here, we show that the tyrosine phosphatase PTP4A1 is highly expressed in fibroblasts from patients with SSc. PTP4A1 and its close homolog PTP4A2 are critical promoters of TGFβ signaling in primary dermal fibroblasts and of bleomycin-induced fibrosis in vivo. PTP4A1 promotes TGFβ signaling in human fibroblasts through enhancement of ERK activity, which stimulates SMAD3 expression and nuclear translocation. Upstream from ERK, we show that PTP4A1 directly interacts with SRC and inhibits SRC basal activation independently of its phosphatase activity. Unexpectedly, PTP4A2 minimally interacts with SRC and does not promote the SRC–ERK–SMAD3 pathway. Thus, in addition to defining PTP4A1 as a molecule of interest for TGFβ-dependent fibrosis, our study provides information regarding the functional specificity of different members of the PTP4A subclass of phosphatases.

[1] Division of Rheumatology, Allergy and Immunology, Department of Medicine, University of California San Diego, 9500 Gilman Dr, La Jolla, CA 92093, USA. [2] Division of Cellular Biology, La Jolla Institute for Allergy and Immunology, 9420 Athena Circle, La Jolla, CA 92037, USA. [3] Department of Medicinal Chemistry and Molecular Pharmacology, Purdue University, 610 Purdue Mall, West Lafayette, IN 47907, USA. [4] Rheumatology Division, Department of Biotechnological and Applied Clinical Sciences, University of L'Aquila, Via Giovanni di Vincenzo, 16/B, 67100 L'Aquila, Italy. [5] Scleroderma Center and Jefferson Institute of Molecular Medicine, 9035 Golden Sunset Ln, Springfield, VA 22153, USA. [6] Division of Rheumatology, Department of Medicine, University of California San Francisco, 505 Parnassus Ave, San Francisco, CA 94143, USA. [7] Core Microscopy Facility, The Scripps Research Institute, 10550 N Torrey Pines Rd, La Jolla, CA 92037, USA. [8] Department of Genetics, Perelman School of Medicine, University of Pennsylvania, Philadelphia, PA 19104, USA. Correspondence and requests for materials should be addressed to N.B. (email: nbottini@ucsd.edu)

Systemic sclerosis (SSc) is a multi-system autoimmune connective tissue disorder that leads to fibrosis of the skin and internal organs, resulting in substantial patient morbidity and mortality. At present, SSc is classified in two forms, the limited cutaneous (lcSSc) and the diffuse cutaneous (dcSSc), which differ in the extent of cutaneous involvement[1]. Similar to other fibrotic diseases, the pathogenesis of fibrosis in SSc involves activation of fibroblasts, which leads to excessive deposition of extracellular matrix components and differentiation of α-smooth muscle actin (αSMA) expressing myofibroblast[2, 3]. Fibroblasts and myofibroblasts are considered important target cells for therapeutic interventions aimed at preventing and reversing fibrosis in SSc and other fibrotic diseases[2, 3]. Activation of the transforming growth factor β (TGFβ) pro-fibrotic signaling pathway in fibroblasts is believed to have an important function in SSc and fibrosis in general[4, 5]. Fresolimumab, an antibody able to block signaling through all isoforms of TGFβ, is currently in a clinical trial for SSc[6].

Inhibitors of protein tyrosine kinases are also being extensively investigated as potential anti-fibrotic agents in SSc[7]. For example, nintedanib, a protein tyrosine kinase inhibitor approved by the FDA for idiopathic pulmonary fibrosis[8], has shown promising results in experimental models of SSc[9]. By contrast, the potential profibrotic or antifibrotic effects of protein tyrosine phosphatases (PTP), enzymes that counterbalance protein tyrosine kinases in signal transduction by dephosphorylating phosphotyrosine residues, are mostly unknown, both in SSc and other fibrotic disorders. Aside from pioneer studies showing that the phosphoinositide phosphatase PTEN displays reduced expression

in SSc dermal fibroblasts (DF)[4, 10], only the oxidative inhibition of the tyrosine-specific phosphatase PTP1B has been shown to be involved in promoting platelet-derived growth factor signaling in SSc fibroblasts[11]. Some PTPs have been reported to modulate TGFβ signaling: for example, SHP-2 and PTPRA enhance TGFβ signaling, although their mechanism of action has not yet been established[12, 13].

In this study, we assess the expression of all PTPs[14] in DFs of patients with dcSSc, in whom the occurrence of fibrosis is generally early and rapidly progressive, finding that PTP4A1 is overexpressed in dermal SSc fibroblasts. PTP4A1 belongs to a sub-class of three prenylated PTP (PTP4A1/2/3), which promote growth and migration of tumor cells through mechanisms that are not understood but probably include regulation of growth factor signaling. The three enzymes are highly homologous to each other, and, although they display different tissue-expression pattern and different substrate specificity in vitro, it is unclear whether they have different functions within the same cell types[15–17]. The PTP4A sub-class of PTPs has very low activity in vitro and robust substrates have not been reported for these enzymes. Some of the functions of these enzymes are probably exerted through protein–protein interaction. Indeed, PTP4A1 interacts with, and activates, the p115 Rho GTPase-activating protein (RhoGAP)[18]. All three PTP4A enzymes have also been identified as important enhancers of intracellular magnesium levels, through their physical interaction with the CNNM family of transmembrane magnesium channels[19, 20].

Here, we show that PTP4A1 promotes TGFβ signaling in human DFs and exacerbates experimental fibrosis in mice. In

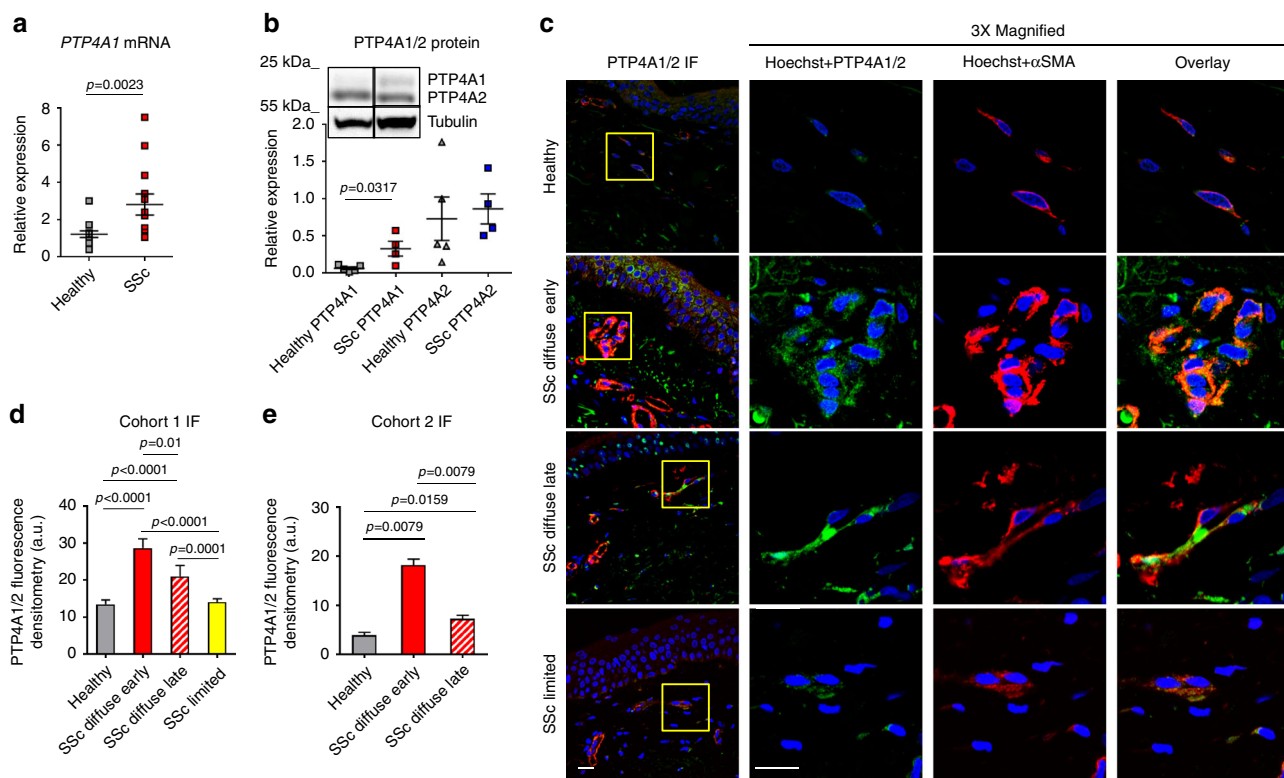

**Fig. 1** PTP4A1 is overexpressed in SSc DFs. **a** Mean ± SEM of *PTP4A1* mRNA expression measured in DF lines from 13 healthy donors or patients with SSc. **b** Mean ± SEM of densitometric scan expression plus representative immunoblotting for PTP4A1/2 (upper bands) normalized to αtubulin (lower bands) in five healthy and four SSc DF lines. Images are from distinct membranes. **c** PTP4A1/2 (green signal) and αSMA (red signal) IF in skin sections from healthy or SSc diffuse early, diffuse late, and limited donors with Hoechst nuclear staining (blue signal). PTP4A1/2-αSMA co-localization signal appears orange in the images. Right panels show magnified area from yellow squares in left panels. **d**, **e** Mean ± SEM of PTP4A1/2 fluorescence densitometry signal in αSMA positive skin cells from two cohorts: the first one **d** including six healthy or five SSc diffuse early, six diffuse late and six limited donors; the second one **e** including five healthy or five SSc early and five late donors. Mann–Whitney test. Scale bars, 50 μm

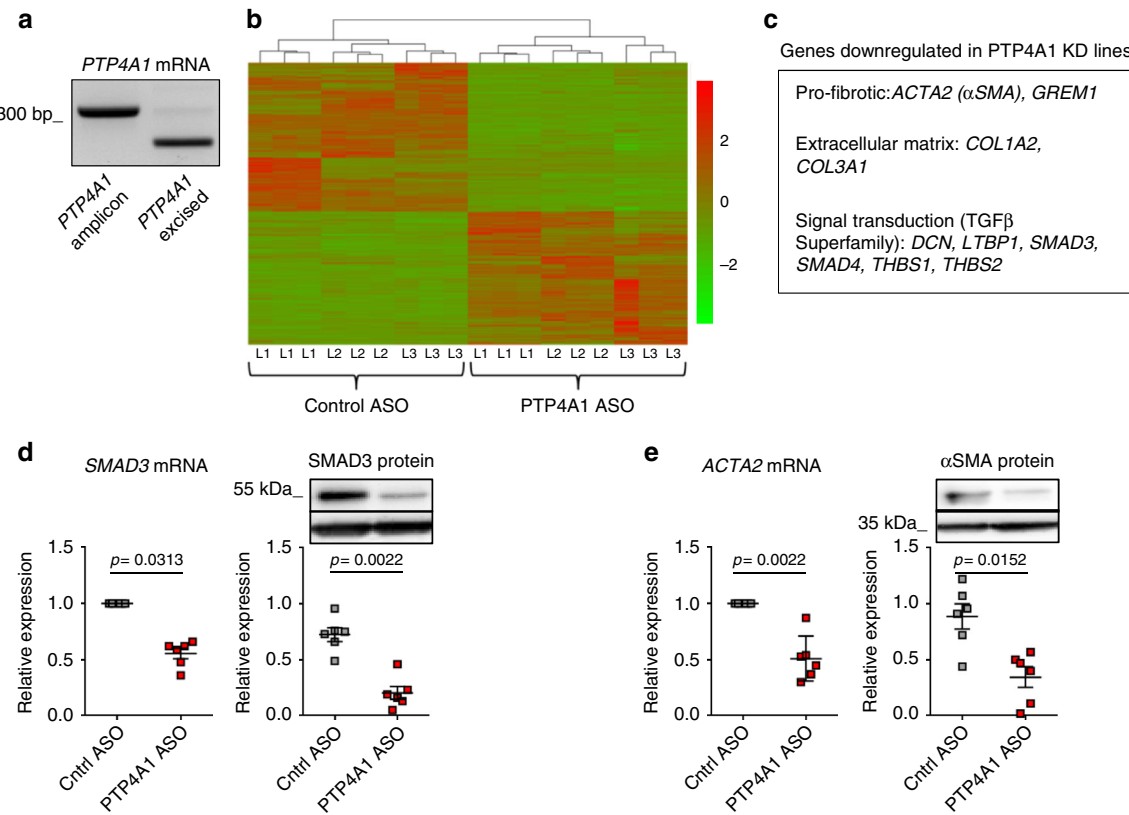

**Fig. 2** Silencing of PTP4A1 downregulates pro-fibrotic genes in NHDF. **a** Agarose gel with RT-PCR of *PTP4A1* mRNA from NHDF lines treated with control ASO (left lane) or with PTP4A1 ASO (right lane). **b** The heat map shows gene expression levels of three different NHDF lines (L1–3) treated with control ASO or PTP4A1 ASO. NGS was performed in triplicate for each NHDF line. **c** List of human pro-fibrotic genes downregulated in PTP4A1 KD NHDF lines, after DESeq analysis. **d**, **e** Left graphs show mean ± SEM of *SMAD3* and *ACTA2* mRNA expression measured in six different NHDF lines treated with PTP4A1 ASO and normalized to same lines treated with control ASO. Right graphs show mean ± SEM of densitometric scan expression plus representative immunoblotting for SMAD3 and αSMA (upper bands) normalized to GAPDH (lower bands) in six different NHDF lines treated with control ASO or PTP4A1 ASO. Wilcoxon test (**d**, **e** left) or Mann–Whitney test (**d**, **e** right)

human primary fibroblasts, we show that PTP4A1 sustains extracellular regulated kinase (ERK)-dependent expression of the important TGFβ mediator mothers against decapentaplegic homolog 3 (SMAD3). In cells with PTP4A1 knockdown, reduced SMAD3 expression and ERK activation correlates with reduced SRC half-life and overall activity. We propose a molecular mechanism in which PTP4A1 binds directly to SRC and protects it from excessive degradation and functional inhibition. Intriguingly, PTP4A2 also promotes TGFβ signaling in human fibroblasts, but only minimally affects the activation of the SRC–ERK–SMAD3 pathway. Thus, we also provide initial evidence of target and functional selectivity between members of the PTP4A sub-class co-expressed in a specific cell type.

## Results

**PTP4A1 is overexpressed in SSc DFs.** We first assessed the mRNA levels of all PTPs (109 genes) in four DF lines from patients with diffuse SSc (dcSScDF) and found that 18 PTPs—belonging to all PTP subclasses—displayed mRNA levels comparable to or higher than the housekeeping DNA-directed RNA polymerase II subunit RPB1 (*POLR2A*) gene (Supplementary Fig. 1). Next, we compared the mRNA expression of these 18 PTPs in the four dcSScDF and in five DF lines from normal human subjects (NHDF). We found that the mRNAs encoding for PTP4A1 and PTP4A2 were highly expressed in both NHDF and dcSScDF. However, *PTP4A1* displayed up to five times overexpression in dcSScDF vs. NHDF fibroblasts, while no difference in *PTP4A2*

expression was detected between the two groups. PTP4A1 and PTP4A2 are close homologs (87% identity in primary structure) and are both overexpressed in tumors and tumor metastasis[15, 17]. We thus extended the study of *PTP4A1* and *PTP4A2* to nine additional dcSScDF and eight NHDF lines. In aggregate, *PTP4A1* displayed two times overexpression in dcSScDF lines derived from 13 SSc patients, when compared with 13 NHDF lines, while no differences were found in *PTP4A2* levels (Fig. 1a). We next assessed the expression of PTP4A1 and PTP4A2, at the protein level, in DF lysates. Owing to the lack of an antibody that displays specificity for PTP4A1 vs. PTP4A2, we distinguished the two proteins on denaturing polyacrylamide gels, since PTP4A2 resolves at a slightly lower molecular weight than PTP4A1. As shown in Fig. 1b, we found significantly increased expression of PTP4A1 but not PTP4A2 in dcSScDF compared to NHDF lines.

To confirm that PTP4A1 is overexpressed in affected SSc skin DFs, we performed immunofluorescence (IF) assessment of PTP4A1 and PTP4A2 expression in dermal αSMA-positive myofibroblasts, comparing biopsies from patients with early or late dcSSc and limited SSc with biopsies from healthy controls. As shown in Fig. 1c and d, myofibroblasts in the dermis of American patients with early dcSSc (< 3 years from first non-Raynaud manifestation) displayed a significantly increased PTP4A1/PTP4A2 signal when compared to those from late dcSSc (> 3 years from first non-Raynaud manifestation), limited SSc and normal controls. To confirm our observations we analyzed biopsies from a second cohort of 10 early dcSSc patients and 5 healthy donors from the Italian population (Fig. 1e and Supplementary

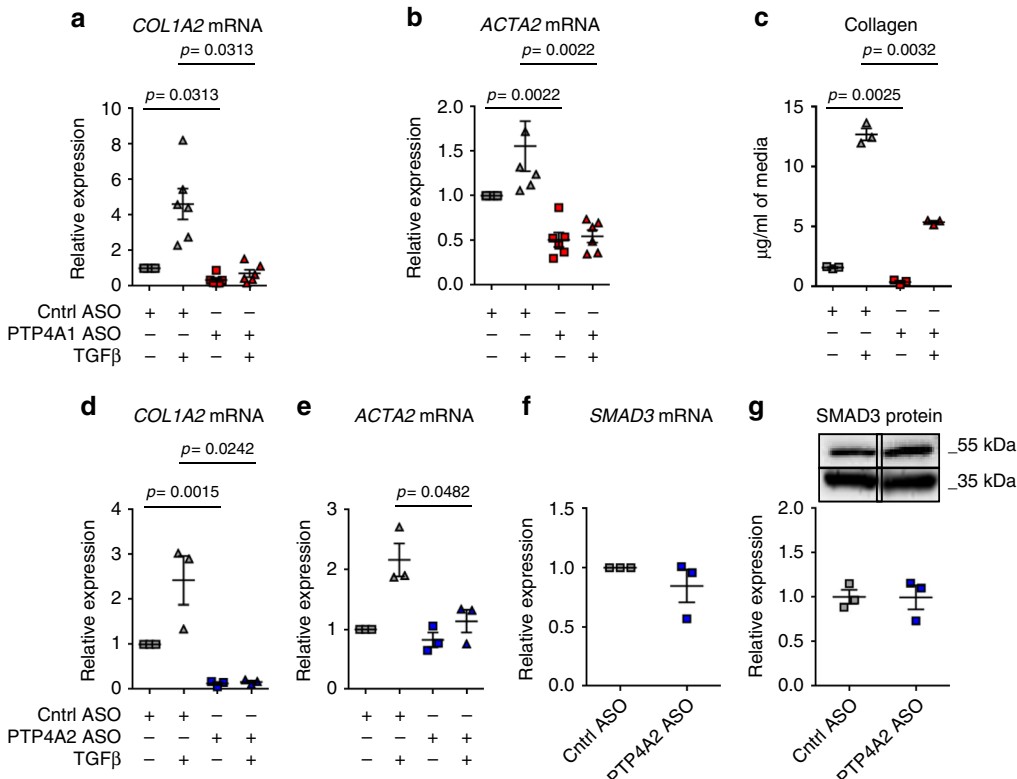

**Fig. 3** PTP4A1 and PTP4A2 silencing inhibits TGFβ signalings. **a**, **b** Mean ± SEM of *COL1A2* and *ACTA2* mRNA expression measured in six different NHDF lines treated with PTP4A1 ASO, stimulated with TGFβ1 and normalized to the same lines treated with control ASO. **c** Mean ± SEM of collagen concentration measured in three different NHDF lines treated with control or PTP4A1 ASO and stimulated with TGFβ1. **d**, **e** Mean ± SEM of *COL1A2* and *ACTA2* mRNA expression measured in three different NHDF lines treated with PTP4A2 ASO, stimulated with TGFβ1 and normalized to the same lines treated with control ASO. **f** Mean ± SEM of *SMAD3* mRNA expression measured in three different NHDF lines treated with PTP4A2 ASO and normalized to the same lines treated with control ASO. **g** Mean ± SEM of densitometric scan expression plus representative immunoblotting for SMAD3 (upper bands) normalized to GAPDH (lower bands) in three different NHDF lines treated with control ASO or PTP4A2 ASO. Images are from different lanes on the same membrane. Wilcoxon test (**a**, **b**), Welch's *t*-test (**c**), or Paired *t*-test (**d**, **e**)

Fig. 2A). In this Italian cohort, we confirmed the overexpression of PTP4A1/PTP4A2 in dermal myofibroblasts of dcSSc biopsies when compared with healthy biopsies. Furthermore, myofibroblasts from the five biopsies collected < 3 year from first non-Raynaud manifestation showed increased expression of PTP4A1/PTP4A2 when compared to biopsies collected > 3 year from first non-Raynaud manifestation. We conclude that PTP4A1 is significantly overexpressed in DFs of patients with dcSSc.

Since TGFβ is considered a key cytokine activating the pro-fibrotic program in dcSSc DFs, we examined whether PTP4A1 expression is upregulated in DFs in response to TGFβ stimulation. As shown in Supplementary Fig. 2B, TGFβ stimulation of NHDF-induced overexpression of *PTP4A1*, but not *PTP4A2*, suggesting that increased TGFβ signaling might underlie the selective overexpression of PTP4A1 in SSc fibroblasts.

**Silencing of *PTP4A1* downregulates profibrotic genes in NHDF**. To understand the role of PTP4A1 in DFs we optimized silencing of *PTP4A1* in NHDF, by using a cell-permeable anti-sense morpholino oligonucleotide (ASO)[21, 22]. ASO treatment enabled nearly complete knockdown of *PTP4A1* (Fig. 2a). Next, using next-generation sequencing (NGS), we determined the transcriptional profile of three NHDF lines treated with PTP4A1 ASO compared to the same lines treated with a control non-targeting ASO (Cntrl ASO). After differential expression calculation, 1886 genes displayed a log2 fold change > 0.5 in expression and an adjusted *p*-value < 0.01 using Benjamini–Hochberg

method in NHDF subjected to silencing of *PTP4A1*. Differentially expressed genes between PTP4A1 ASO-treated NHDF and Cntrl ASO-treated NHDF are represented as a heat map in Fig. 2b. Moreover, we analyzed molecular functions, biological processes, and cellular compartments of differentially expressed genes using the GO-Elite software[23]. We found that extracellular matrix organization and cell proliferation pathways were downregulated in *PTP4A1*-silenced NHDF (Supplementary Fig. 3). A subsequent pairwise comparison of PTP4A1 ASO-treated NHDF vs. Cntrl ASO-treated NHDF using the DESeq software[24] showed that several genes involved in human fibrosis were significantly downregulated in *PTP4A1*-silenced NHDF, including genes encoding collagen chains (*COL1A2* and *COL3A1*) and αSMA (*ACTA2*) (Fig. 2c). We also noticed that PTP4A1 knockdown resulted in a 1.3-fold decrease in the mRNA expression of *SMAD3*, which encodes for a key mediator of pro-fibrotic TFGβ signaling[25]. The downregulation of both *ACTA2* and *SMAD3* was further confirmed at the mRNA and protein levels in six additional NHDF lines, subjected to ASO-mediated silencing of *PTP4A1* (Fig. 2d, e). These results point to a possible role of PTP4A1 in dermal-fibroblast-mediated skin fibrosis. Furthermore, the downregulation of SMAD3 expression in *PTP4A1*-silenced NDHF also suggests a possible role of PTP4A1 in modulating pro-fibrotic TGFβ signaling in DFs.

**PTP4A1 and PTP4A2 promote TGFβ signaling in fibroblasts**. In order to evaluate the effect of PTP4A1 on TGFβ signaling in

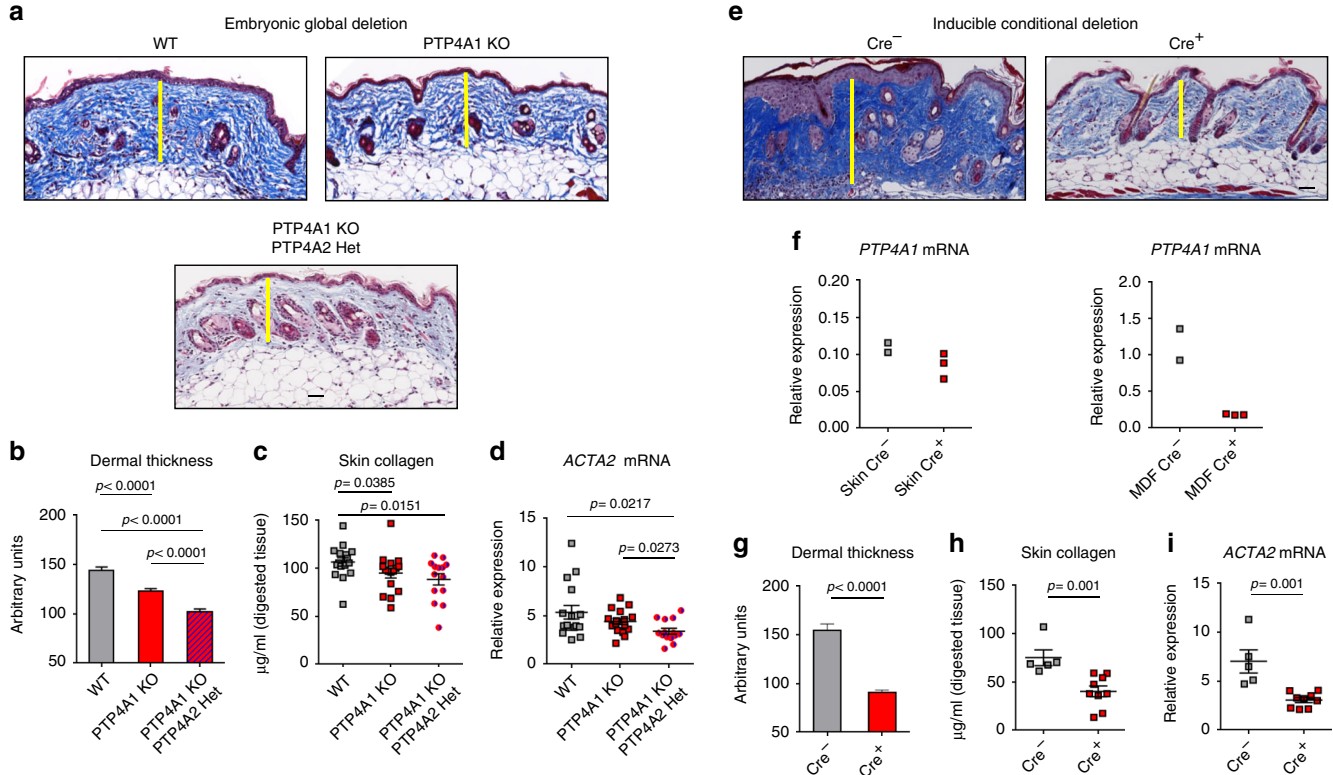

**Fig. 4** PTP4A1 silencing protects mice from dermal fibrosis. Masson's trichrome-stained skin sections from bleomycin-treated mice. Images are representative of 16 WT, 16 PTP4A1 KO, and 14 PTP4A1 KO/PTP4A2 Het mice (**a**), or 5 COL1A1Cre⁻/PTP4A1^fl/fl and 9 COL1A1Cre⁺/PTP4A1^fl/fl tamoxifen-induced mice (**e**). Graphs show mean ± SEM dermal thickness (**b**, **g**, see yellow bars in **a** and **e**), skin collagen concentration (**c**, **h**), and *ACTA2* mRNA relative expression (**d**, **i**). **f** *PTP4A1* mRNA relative expression in whole skin (left graph) or in isolated mouse dermal fibroblasts (right graph) from two COL1A1Cre⁻/PTP4A1^fl/fl or three COL1A1Cre⁺/PTP4A1^fl/fl mice. Mann–Whitney test. Scale bar, 50 μm

DFs, we stimulated with TGFβ1 both PTP4A1 ASO-treated NHDF and Cntrl ASO-treated NHDF. We found a significant reduction in basal and/or TGFβ-stimulated expression of *COL1A1*, *COL1A2*, *ACTA2*, *CTGF* as well as collagen secretion in *PTP4A1*-silenced NHDF (Fig. 3a–c and Supplementary Fig. 4). Since PTP4A2 is highly homologous to PTP4A1 and is expressed at comparable levels in DFs, we assessed whether the two phosphatases play similar roles on TGFβ signaling in NHDF lines. NHDF were treated with Cntrl ASO or ASO designed to silence *PTP4A2* (PTP4A2 ASO). The PTP4A2 ASO enabled nearly complete knockdown of *PTP4A2* (Supplementary Fig. 5A). Similar to *PTP4A1*-silenced fibroblasts, in *PTP4A2*-silenced fibroblasts we also found a significant reduction in basal and TGFβ-stimulated expression of *COL1A2*, and in TGFβ-stimulated expression of *ACTA2* (Fig. 3d, e). However, there was no evidence of reduction in SMAD3 expression at the mRNA and protein levels (Fig. 3f, g).

ASOs are designed against pre-mRNA splice junctions and have a negligible chance of off-target silencing[22]. Nevertheless, to confirm that the effects of PTP4A1 ASO were specifically due to *PTP4A1* silencing, we designed a second ASO targeting a different splicing junction on the *PTP4A1* pre-mRNA (PTP4A1 ASO2), and confirmed comparable silencing efficiency to PTP4A1 ASO (Supplementary Fig. 5A). Supplementary Fig. 5B shows that knockdown of all ASOs in NHDF was confirmed at protein level. Next, we incubated NHDF lines with Cntrl ASO or PTP4A1 ASO2. Similar to what observed in PTP4A1 ASO-treated NDHF lines, PTP4A1 ASO2-treated NHDF lines displayed reduced SMAD3 expression (Supplementary Fig. 5C) and a significant reduction in basal and TGFβ-stimulated *COL1A2* expression (Supplementary Fig. 5D).

Since lung fibrosis is a frequent and potential life threatening complication of SSc, we sought to determine whether promotion of TGFβ signaling is a conserved function of PTP4A1 between dermal and lung fibroblasts. Thus, we incubated lung fibroblast lines from healthy donors (NHLF) with Cntrl or PTP4A1 ASO. *PTP4A1* silencing in NHLF lines (Supplementary Fig. 6A) resulted in reduced basal and TGFβ-stimulated expression of *COL1A2* and significantly decreased *SMAD3* expression (Supplementary Fig. 6B–D), similar to those observed in NHDF lines.

We conclude that both PTP4A1 and PTP4A2 promote TGFβ pro-fibrotic signaling in human dermal and lung fibroblasts. However, the lack of redundancy between the two phosphatases and the selective promotion of SMAD3 expression by PTP4A1 suggest that PTP4A1 and PTP4A2 modulate fibroblast TGFβ signaling through non-overlapping mechanisms of action.

**PTP4A1 and PTP4A2 knockout (KO) protects mice from dermal fibrosis**. In order to understand whether PTP4A1 plays a role in fibrosis in vivo, we generated a mouse strain carrying a global deletion of *PTP4A1*. By analyzing more than 500 pups produced by *PTP4A1* heterozygous mating, we found that the distribution of wild-type (WT) and mutant alleles did not exactly match the expected Mendelian ratios. Although a significantly reduced number of homozygous global *PTP4A1* KO mice were identified in the offspring, these *PTP4A1* KO mice displayed normal fertility and no clinically evident pathology was observed in follow-up to the age of 24 months. Since both PTP4A1 and PTP4A2 promote pro-fibrotic TGFβ signaling in fibroblasts, we also sought to generate PTP4A1/PTP4A2 double KO mice by crossing the PTP4A1 heterozygous mice with the PTP4A2 KO mice described

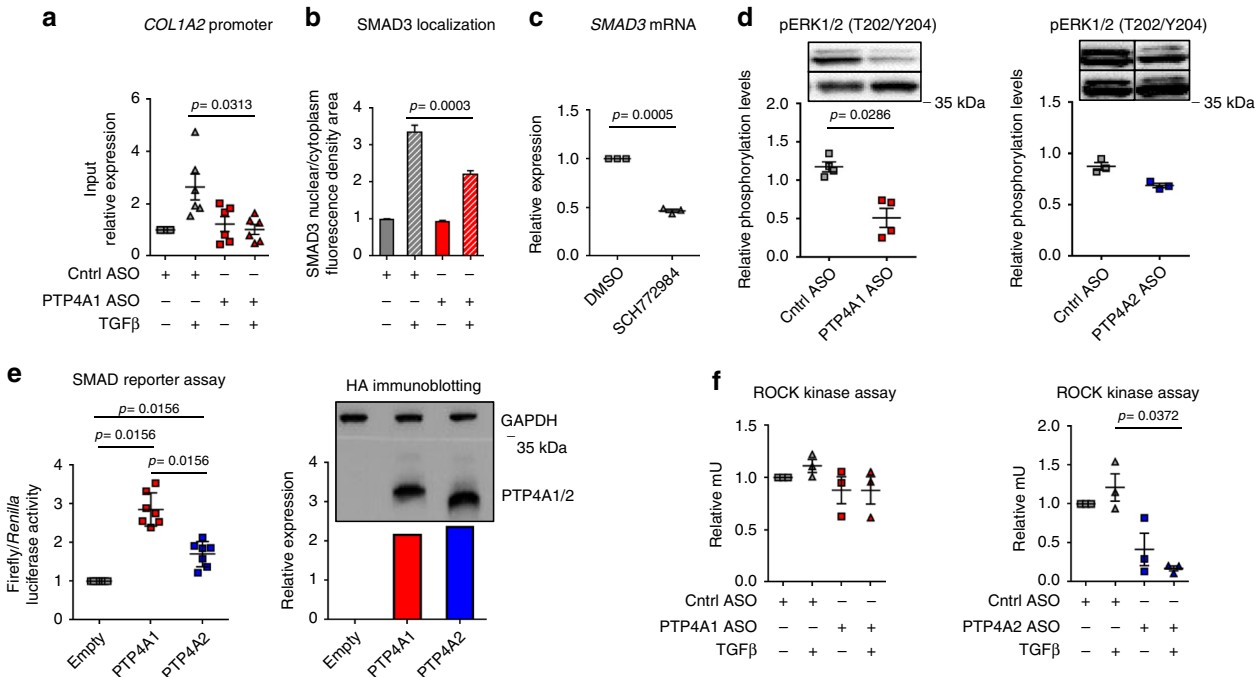

**Fig. 5** PTP4A1 influences TGFβ pro-fibrotic signaling through SMAD3. **a** Mean ± SEM of COL1A2 promoter expression in six different NHDF lines treated with PTP4A1 ASO, stimulated with TGFβ1 and normalized to the same lines treated with control ASO. pSMAD3 antibody was used to perform the ChIP. **b** Mean ± SEM of the ratio between SMAD3 nuclear and cytoplasmatic fluorescence densitometry in three cell lines treated with control ASO and PTP4A1 ASO and then stimulated with TGFβ1. **c** Mean ± SEM of SMAD3 mRNA expression measured by qPCR in three NHDF lines treated with 50 μm SCH772984 ERK inhibitor and normalized to the same lines treated with DMSO. **d** Mean ± SEM of densitometric scan expression plus representative immunoblotting for pERK1/2 (T202/Y204) (upper bands) normalized to ERK2 (lower bands) in four different NHDF lines treated with control ASO, PTP4A1 ASO (left graph) or PTP4A2 ASO (right graph, three different NHDF lines). Images are from different lanes on the same membrane. **e** Left graph shows mean ± SEM of relative ratio of firefly/Renilla luciferase signal of HEK 293T cells co-transfected with human PTP4A1 WT or PTP4A2 WT-encoding HA-tag pCDNA4 vectors together with a firefly luciferase SMAD reporter and a control luciferase Renilla vector. Cells were stimulated with TGFβ1. Graph is representative of 6 independent experiments. Right graph shows densitometric scan relative expression plus representative immunoblotting for HA-tagged PTP4A1/2 (lower bands) normalized to GAPDH (upper bands) in HEK 293T cells co-transfected with human HA-tagged PTP4A1 WT or PTP4A2 WT-encoding pCDNA4. **f** Mean ± SEM of ROCK kinase activity in three different NHDF lines treated with PTP4A1 (left graph) or PTP4A2 ASO (right graph), stimulated with TGFβ1 and normalized to the same lines treated with control ASO. Wilcoxon test (**a**, **e** left), Mann–Whitney test (**b**, **d**), or paired t-test (**c**, **f**)

previously[26]. Unfortunately, deletion of both PTP4A1 and PTP4A2 is embryonic lethal (data not shown). However, we were able to obtain PTP4A1 KO/PTP4A2 heterozygous (Het) mice. We then subjected WT, PTP4A1 KO or PTP4A1 KO/PTP4A2 Het mice to bleomycin-induced dermal fibrosis, an experimental model that is often used in the study of both early inflammatory and fibrotic stages of SSc[27]. In bleomycin-treated PTP4A1 KO compared to WT mice, there was a significant reduction of dermal thickness, skin collagen concentration, and ACTA2 mRNA expression. Furthermore, the phenotype of protection from fibrosis was even more marked in PTP4A1 KO/PTP4A2 Het mice (Fig. 4a–d). To assess whether the pro-fibrotic effect of PTP4A1 in vivo is mediated by fibroblasts, we took advantage of a mouse model carrying a floxed allele of PTP4A1[10] to generate mice carrying tamoxifen-inducible deletion of PTP4A1 in COL1A1-expressing cells. As shown in Fig. 4e–i Cre⁺ mice, subjected to postnatal tamoxifen treatment, displayed almost complete deletion of PTP4A1 in DFs, and, consistent with our observations in constitutive global PTP4A1 KO mice, they were markedly protected from bleomycin-induced skin fibrosis. We conclude that PTP4A1 and PTP4A2 promote skin fibrosis in vivo.

**PTP4A1 promotes TGFβ signaling through ERK–SMAD3 pathway.** We next sought to understand the mechanism of action of PTP4A1 in modulation of NHDF TGFβ signaling. We first

assessed whether promotion of SMAD3 expression by PTP4A1 results in enhanced SMAD3 occupancy on TGFβ-dependent pro-fibrotic genes. Phospho-SMAD3 (S423/S425) chromatin immunoprecipitation (CHIP) in NHDF treated with PTP4A1 ASO showed significantly reduced TGFβ-induced SMAD3 occupancy on the COL1A2 promoter compared with Cntrl ASO-treated cells (Fig. 5a). IF assessment of SMAD3 localization, in the same cells, confirmed overall reduced levels of SMAD3 (Supplementary Fig. 7A). In addition, we also found a reduced nuclear-cytoplasmic SMAD3 ratio in TGFβ-stimulated PTP4A1 ASO-treated NHDF lines when compared with Cntrl ASO-treated NHDF lines (Fig. 5b and Supplementary Fig. 7A). These data suggest that reduced TGFβ-induced SMAD3 occupancy of target loci in PTP4A1-silenced fibroblasts is due to a combination of decreased expression of SMAD3 and diminished TGFβ-induced nuclear translocation or retention. Phosphorylation of SMAD3 on S423/S425 is necessary for its cytoplasmic-nuclear translocation[25]. However, NHDF lines, subjected to PTP4A1 silencing, did not display any significant difference in TGFβ-induced phosphorylation of SMAD3 S423/425 (Supplementary Fig. 7B). SMAD3 expression has been reported to be dependent on ERK1/2 mitogen-activated protein kinase (MAPK) pathway activation in human alveolar type II epithelial adenocarcinoma cell line (A549)[28]. In fibroblasts, phosphorylation of SMAD3 by ERK1/2 on S204 also promotes its COL1A2-activating function[29]. Reportedly, all three prenylated PTPs promote activation of

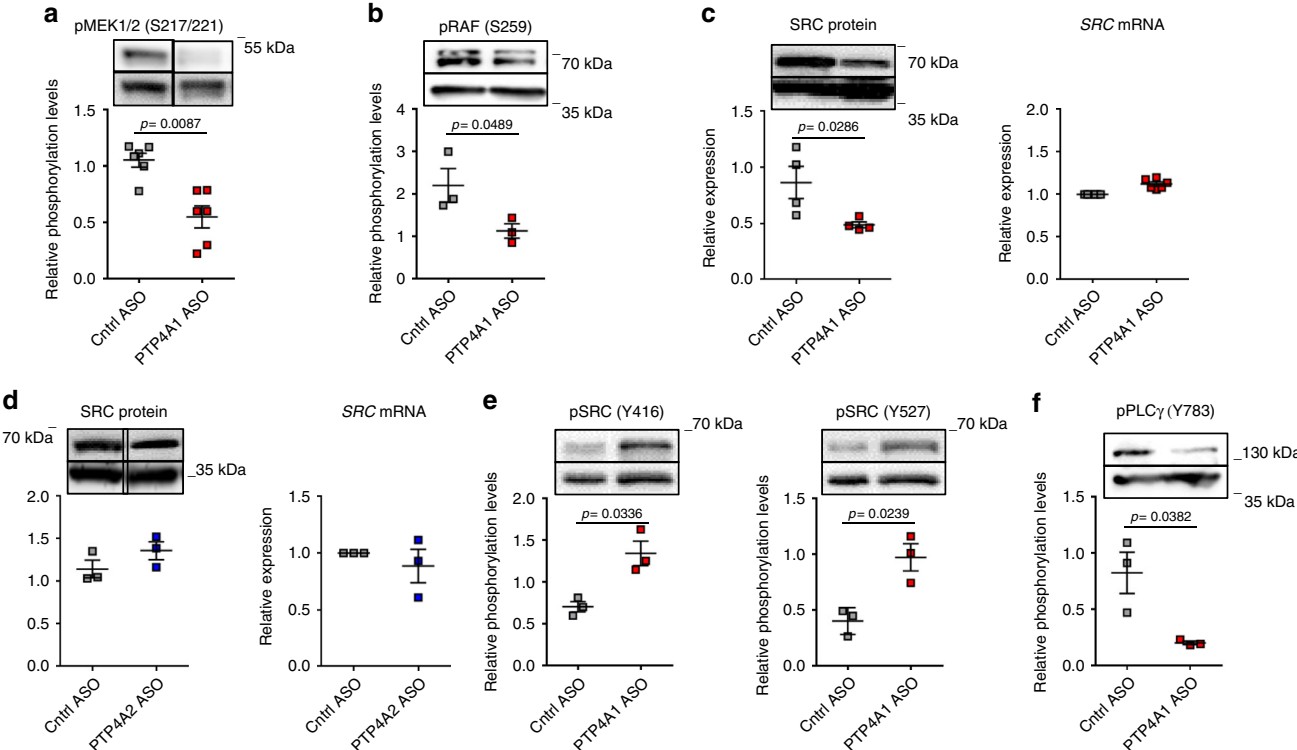

**Fig. 6** PTP4A1 enhances SRC half-life and activity in NHDF. **a**, **b** Mean ± SEM of densitometric scan expression plus representative immunoblotting for pMEK1/2 (S217/221) or pRAF (S259) (upper bands) normalized to MEK1/2 or GAPDH, respectively (lower bands) in three different NHDF lines treated with control or PTP4A1 ASO. Images are from different lanes on the same membrane. **c**, **d** Left graphs show mean ± SEM of densitometric scan expression plus representative immunoblotting for SRC (upper bands) normalized to GAPDH (lower bands) in four different NHDF lines treated with control ASO, PTP4A1 ASO, or PTP4A2 ASO (three different NHDF lines). Right graphs show mean ± SEM of SRC mRNA expression measured in six different NHDF lines treated with PTP4A1 ASO or three different NHDF lines treated with PTP4A2 ASO and normalized to the same lines treated with control ASO. Images are from different lanes on the same membrane. **e**, **f** Mean ± SEM of densitometric scan expression plus representative immunoblotting for pSRC (Y416) (**e**, left graph, upper bands), pSRC (Y527) (**e**, right graph, upper bands), or pPLCγ (Y783) (**f**, upper bands) normalized to SRC (lower bands in **e**) or GAPDH (lower bands in **f**) in three different NHDF lines treated with control or PTP4A1 ASO. Mann–Whitney test (**a**) or Welch's t-test (**b**, **c**, **e**, **f**)

ERK1/2 in cancer cells[30]. Thus, we hypothesized that reduced basal activation of ERK1/2 might explain the reduced basal expression and TGFβ-induced nuclear translocation of SMAD3 in *PTP4A1*-silenced NHDF lines. In order to test this hypothesis, we first assessed whether SMAD3 expression and its TGFβ-induced nuclear translocation are ERK-dependent in NHDF lines. Figure 5c and Supplementary Fig. 8A show that treatment of NHDF with the selective ERK1/2 inhibitor SCH772984[31] resulted in significantly reduced SMAD3 mRNA level and reduced TGFβ-induced SMAD3 nuclear translocation, when compared to DMSO-treated cells. Next, we assessed whether PTP4A1 and PTP4A2 promote ERK1/2 activity in NHDF lines. Figure 5d shows that NHDF lines, treated with PTP4A1 ASO, displayed significantly reduced basal phosphorylation of ERK1/2 on T202/Y204, when compared with Cntrl ASO-treated NDHF lines. However, consistent with the observed lack of differences in SMAD3 expression in PTP4A2 ASO-treated NHDF lines, *PTP4A2* silencing only caused a small non-significant trend in reduced ERK1/2 phosphorylation. Consistent with the hypothesis that PTP4A1 promotes SMAD3 cytosolic-nuclear shuttling via ERK activation, we did not observe any evidence of altered TGFβ-induced nuclear translocation of SMAD3 in *PTP4A2*-silenced NHDF lines (Supplementary Fig. 8B). Overall, our data suggest that promotion of TGFβ signaling by PTP4A1 in NHDF is at least in part mediated by enhanced basal activation of ERK1/2, which results in enhanced expression of SMAD3, and primes it for enhanced TGFβ-induced nuclear translocation. Consistent with this hypothesis, DF lines from SSc patients displayed significant

increased ERK1/2 activity when compared to NHDF (Supplementary Fig. 8C). Intriguingly, this function of PTP4A1 is not or only minimally shared by PTP4A2 in NHDF lines, despite the high homology between the two phosphatases and the fact that both PTP4A1 and PTP4A2 promote TGFβ signaling in NHDF lines. To gather further support to the observed different function exerted by PTP4A1 vs. PTP4A2 in TGFβ signaling, we assessed whether overexpression of PTP4A1 enhances the activation of a SMAD-dependent luciferase reporter in HEK 293T cells. This reporter assay has been described to be ERK-dependent[28] and we confirmed that it could be inhibited by cell treatments with SCH772984 (Supplementary Fig. 8D). Thus, we co-transfected HEK 293T cells with the SMAD luciferase reporter and human pCDNA4-encoded PTP4A1 or PTP4A2. After TGFβ stimulation, luciferase activity was enhanced 2.7 times in cells expressing PTP4A1, but only 1.6 times in cells expressing PTP4A2, despite the transfected cells displayed equal expression levels of the two phosphatases (Fig. 5e).

PTP4A1 and PTP4A3 have been reported to promote activation of Rho and of its major effector Rho-associated protein kinase (ROCK) in cancer cells[32]. ROCK activity is increased in pulmonary fibrosis[33]. ROCK also promotes pro-fibrotic TGFβ signaling in DF and inhibition of ROCK protects bleomycin-treated mice from the development of dermal fibrosis[34]. Thus, we assessed whether increased activation of ROCK contributes to the PTP4A1-mediated enhancement of TGFβ signaling in NHDF lines. As shown in Fig. 5f, while *PTP4A1* silencing had no effect on ROCK activity in NHDF lines, *PTP4A2* silencing led to a

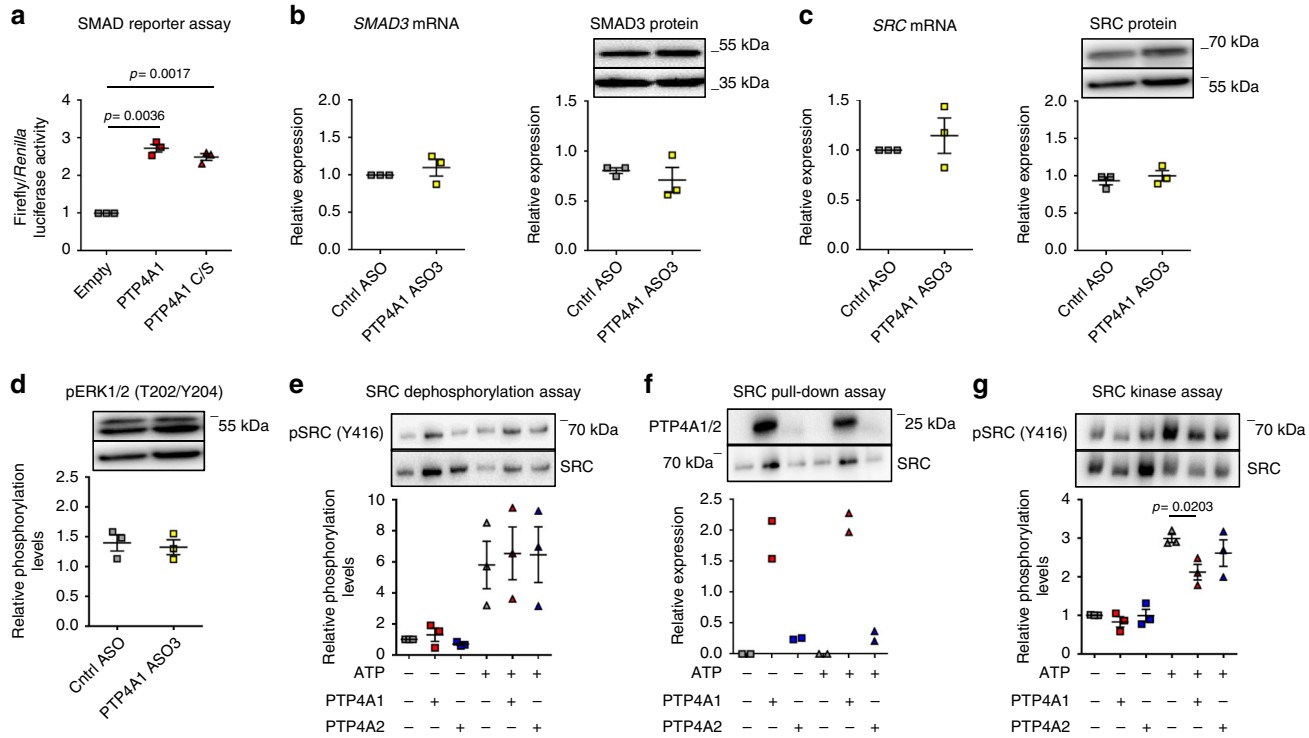

**Fig. 7** PTP4A1 inhibits the basal autophosphorylation of SRC. **a** Mean ± SEM of relative ratio of firefly/*Renilla* luciferase signal of HEK 293T cells co-transfected with human HA-tagged PTP4A1 WT or PTP4A1 C104S (phosphatase dead)-encoding pCDNA4 vector together with a firefly luciferase SMAD reporter and a control *Renilla* luciferase vector. Cells were stimulated with TGFβ1. Graph is representative of three independent experiments. **b**–**d** Mean ± SEM of *SMAD3* or *SRC* mRNA expression or densitometric scan relative expression plus representative immunoblotting for SMAD3, SRC, pERK1/2 (T202/Y204) (upper bands) normalized to GAPDH or αtubulin (lower bands) in three different NHDF lines treated with control ASO or PTP4A1 ASO3. **e** Mean ± SEM of densitometric scan expression plus representative immunoblotting for pSRC (Y416) (upper bands) normalized to SRC (lower bands) in three different in vitro SRC kinase assay followed by incubation with PTP4A1/PTP4A2. **f** Densitometric scan expression plus representative immunoblotting for PTP4A1/2 (upper bands) normalized to SRC (lower bands) in two different in vitro SRC pull-down assay in presence of PTP4A1 or PTP4A2. **g** Mean ± SEM of densitometric scan expression plus representative immunoblotting for pSRC (Y416) (upper bands) normalized to SRC (lower bands) in three different in vitro SRC kinase assay carried out in presence of PTP4A1/PTP4A2. Paired *t*-test

dramatic inhibition of ROCK activity in TGFβ-stimulated NHDF lines.

Since PTP4A1 inhibits CNNM function leading to increased intracellular Mg levels in HeLa, COS7, and HEK 293 cells[35], we also assessed whether regulation of CNNM channel physiology contributes to the action of PTP4A1 in TGFβ signaling. Since HEK 293 cells almost exclusively express CNNM4 channels[35], we examined the effect of knockdown of CNNM4 in HEK 293 cells on SMAD reporter activation by TGFβ. As shown in Supplementary Fig. 9A, knockdown of CNNM4 did not affect TGFβ-mediated reporter activation, suggesting that the mechanism of action of PTP4A1 in TGFβ signaling is unlikely mediated by decreased CNNM4 function. In line with this model, reporter assays carried out in Mg-free media also showed that while Mg was necessary for full reporter activation, reporter enhancement by PTP4A1 was unaffected by Mg deficiency (Supplementary Fig. 9B). Taken together, our data suggest that promotion of TGFβ-induced pro-fibrotic gene expression by PTP4A1 occurs through basal enhancement of ERK activation, which in turn boosts transcription and nuclear translocation/retention of SMAD3, resulting in promotion of canonical TGFβ signaling. Additionally, our data suggest that PTP4A1 and PTP4A2 regulate different aspects of TGFβ signaling in NHDF signaling.

**PTP4A1 enhances SRC half-life in NHDF lines**. To understand how PTP4A1 promotes activation of ERK1/2 in NHDF lines, we

first assessed whether PTP4A1 affects the activation of upstream MAPK kinases and MAPK kinase kinases (MAPKK and MAPKKK) in the ERK pathway[36]. We found that NDHF lines subjected to *PTP4A1* silencing displayed significantly reduced phosphorylation of the MAPKK mitogen-activated protein kinase kinases 1/2 (MEK1/2) on S217/S221 and of the RAF proto-oncogene serine/threonine-protein kinase (RAF) MAPKKK on S259 (Fig. 6a, b). SRC is a major effector of RAF activation through activation of RAS[37]. In cancer cell lines, two groups have separately reported that PTP4A1 promotes SRC autophosphorylation on Y416 and SRC protein expression levels, respectively[38, 39]. However, the molecular basis of SRC regulation by PTP4A1 remains unclear and the two above-mentioned observations are hard to reconcile since autophosphorylation of SRC Y416 induces its degradation through Cul5-mediated ubiquitination[40, 41]. In order to evaluate whether PTP4A1 promotes the activation of the ERK pathway in NHDF lines through an action on SRC, we assessed the expression and the activity of SRC in NHDF lines subjected to *PTP4A1* silencing. Figure 6c and d shows that *PTP4A1* silencing in NHDF lines resulted in a significant reduction of SRC protein expression, but not of *SRC* mRNA levels, while *PTP4A2* silencing in the same cells did not affect SRC expression at all. Next, we examined SRC phosphorylation levels on Y416 and Y527—the latter is an inhibitory site that is specifically phosphorylated by the C-terminal SRC kinase (CSK)—in NHDF lines subjected to *PTP4A1* silencing. We found significantly increased phosphorylation of SRC on both Y416 and

Y527 in PTP4A1 ASO-treated when compared to Cntrl ASO-treated fibroblasts (Fig. 6e). In order to evaluate SRC activity in cells subjected to *PTP4A1* silencing, we assessed phosphorylation of Y783 of phospholipase C gamma 1 (PLCγ)—a specific SRC phosphorylation site[42]. We found that PTP4A1 ASO-treated cells incubated with PTP4A1 ASO displayed reduced PLCγ-Y783 phosphorylation (Fig. 6f). We conclude that PTP4A1 promotes SRC half-life through decreased basal SRC autophosphorylation on Y416. We hypothesize that in cells expressing PTP4A1 the combination of sustained SRC expression and decreased phosphorylation on Y527 results in enhanced SRC basal activity, which in turn enhances SMAD3 expression and nuclear translocation via ERK activation. In line with this hypothesis, we observed that DF lines from SSc patients displayed significant increased SRC protein level when compared to NHDF (Supplementary Fig. 10A). Moreover, treatment of NHDF with the selective SRC inhibitor SU6656 resulted in significantly reduced TGFβ-induced SMAD3 nuclear translocation, when compared to DMSO-treated cells (Supplementary Fig. 10B).

We next sought to understand how PTP4A1 inhibits phosphorylation of SRC on Y416. Since PTP4A1 is a tyrosine phosphatase, an obvious scenario is that PTP4A1 directly dephosphorylates SRC Y416. To test this hypothesis, we first determined whether the catalytic activity of PTP4A1 is necessary for PTP4A1-mediated promotion of TGFβ signaling by measuring the SMAD luciferase reporter activity in HEK 293T expressing HA-tagged human PTP4A1 or its catalytically dead C104S mutant. These experiments showed that expression of PTP4A1 C104S leads to enhanced TGFβ-induced luciferase activity comparable to the expression of the WT enzyme, suggesting that the phosphatase activity of PTP4A1 is dispensable for its TGFβ signaling promoting action (Fig. 7a). In order to define whether in NHDF lines the phosphatase activity of PTP4A1 is dispensable for its effect on SRC half-life and activity, we designed an ASO for in-frame deletion of *PTP4A1* exon 4 (aa 110–135), thus eliminating R110, which is critical for enzyme catalysis, without altering all the other active site residues (Supplementary Fig. 11A). Indeed, we showed that recombinant Δ110-135PTP4A1 (ΔPTP4A1) lacks enzymatic activity in vitro (Supplementary Fig. 11B). The short alpha helix-turn-alpha helix loop encoded by exon 4 is almost completely conserved between PTP4A1 and PTP4A2, thus is unlikely to contribute significantly to the regulation of SRC by PTP4A1. In PTP4A1 ASO3-treated cells, we did not find any significant reduction in SMAD3 or SRC expression (Fig. 7b, c) as well as basal phosphorylation of ERK on T202/Y204 (Fig. 7d), thus indicating that regulation of SRC activity and expression by PTP4A1 do not require PTP4A1 catalytic activity.

Next, we sought to further understand the action of PTP4A1 on SRC phosphorylation through in vitro molecular assays. We first assessed whether recombinant PTP4A1 is able to dephosphorylate SRC subjected to in vitro autophosphorylation on Y416. Figure 7e shows that neither recombinant PTP4A1 or PTP4A2 dephosphorylated SRC Y416. We conclude that, in line with the above-mentioned observations, PTP4A1 is unlikely to directly dephosphorylate SRC on Y416 in NHDF. In the same in vitro assays, we noticed that recombinant PTP4A1 and PTP4A2 formed a physical complex with active SRC, which was persistent after several washes (Fig. 7f and Supplementary Fig. 12A). Complex formation was much more prominent for PTP4A1 than PTP4A2, suggesting that PTP4A1 is capable of a physical interaction with SRC. Importantly, ΔPTP4A1, which retains PTP4A1 function but not its enzymatic activity, also retains PTP4A1 ability to co-precipitate with SRC (Supplementary Fig. 12B). To confirm this observation in a cellular context, we immunoprecipitated both HA-tagged PTP4A1 and PTP4A2

or SRC from lysates of HEK 293T transfected with HA-tagged catalytically active human PTP4A1 or PTP4A2 or catalytically inactive human PTP4A1-C104S or PTP4A2-C101S. In these experiments, we confirmed that PTP4A1 co-precipitates with SRC much more efficiently than PTP4A2 (Supplementary Fig. 12C, D). The observed physical interaction between PTP4A1 and SRC, coupled with lack of direct dephosphorylation of PTP4A1 by SRC, suggests that regulation of SRC autophosphorylation by PTP4A1 might rather occur through a "chaperone-like" protein–protein interaction mechanism. To test this hypothesis, we subjected recombinant SRC to autophosphorylation in an in vitro kinase assay in the presence of recombinant PTP4A1 or PTP4A2. In this assay, only PTP4A1 significantly inhibited SRC autophosphorylation on tyrosine 416 (Fig. 7g). Thus, binding of PTP4A1 to SRC correlates with inhibition of the basal auto-phosphorylation of SRC, suggesting that PTP4A1 is critical to avoid excessive autophosphorylation of SRC, in the absence of acute cell stimuli. A wide range of subcellular localizations, including membranes, cytosol, and nucleus, has been reported for the PTP4A phosphatases[15]. Prenylation is believed to increase their localization at the plasma membrane[43], which could theoretically enhance their access to the active pool of SRC. However, differences in the behavior of PTP4A1 and PTP4A2, in the above-mentioned cellular and in vitro SRC assays, suggest that prenylation is not critical for recruitment of PTP4A1 to SRC, and presumably, for the regulation of TGFβ signaling by PTP4A1. This consideration is experimentally supported by the fact that we did not find any difference between the ability of WT PTP4A1 and its 1–169 mutant, which lacks all prenylation sites[30], to enhance TGFβ-induced SMAD reporter luciferase activity (Supplementary Fig. 13).

## Discussion

Differentiation into αSMA-expressing myofibroblasts and increased collagen deposition are considered key pro-fibrotic alterations of fibroblast function in SSc[2]. TGFβ signaling is believed to significantly contributing as a major driver of the pro-fibrotic phenotype of SSc fibroblasts and anti-TGFβ treatments are being tested in clinical trials with promising results[5, 6]. Molecular signaling alterations in TGFβ and other signaling pathways in SSc fibroblasts and tissues have been object of intense investigation[4, 5, 44, 45]. However, with the exception of early observations on the PTEN—phosphatidylinositol-4,5-bisphosphate 3-kinase pathway[4, 10] and a single report on PTP1B[11]—the role of PTP has remained unaddressed.

Here, we carried out an unbiased survey of PTP mRNA expression in SSc fibroblasts and found that PTP4A1 mRNA and protein are significantly overexpressed in SScDF lines derived from patients in the early stage of the disease, when compared with lines derived from patients at the later stage, patients with limited SSc and healthy controls. Furthermore, the expression of PTP4A1/PTP4A2 was also significantly increased in the skin of the same patients. A few scenarios are envisioned to explain the mechanism of PTP4A1 overexpression in SSc DFs. SSc fibroblast lines display multiple imprinted alterations in signaling and gene expression, including enhanced TGFβ signaling and increased expression of TGFβ receptors[46, 47]. Epigenetic anomalies likely underlie at least in part these imprinted phenotypes of SScDF lines in culture[48]. It is possible that the PTP4A1 locus displays epigenetic alterations in SScDF cells. Alternatively, or in addition, the experimental induction of *PTP4A1* by TGFβ in NHDF lines suggests that increased levels of TGFβ in the pro-fibrotic milieu of SSc skin, together with enhanced endogenous TGFβ sensitivity of SSc fibroblasts, might underlie the overexpression of PTP4A1 in early SSc skin stages and SSc fibroblasts[5, 46, 47]. Since early stages

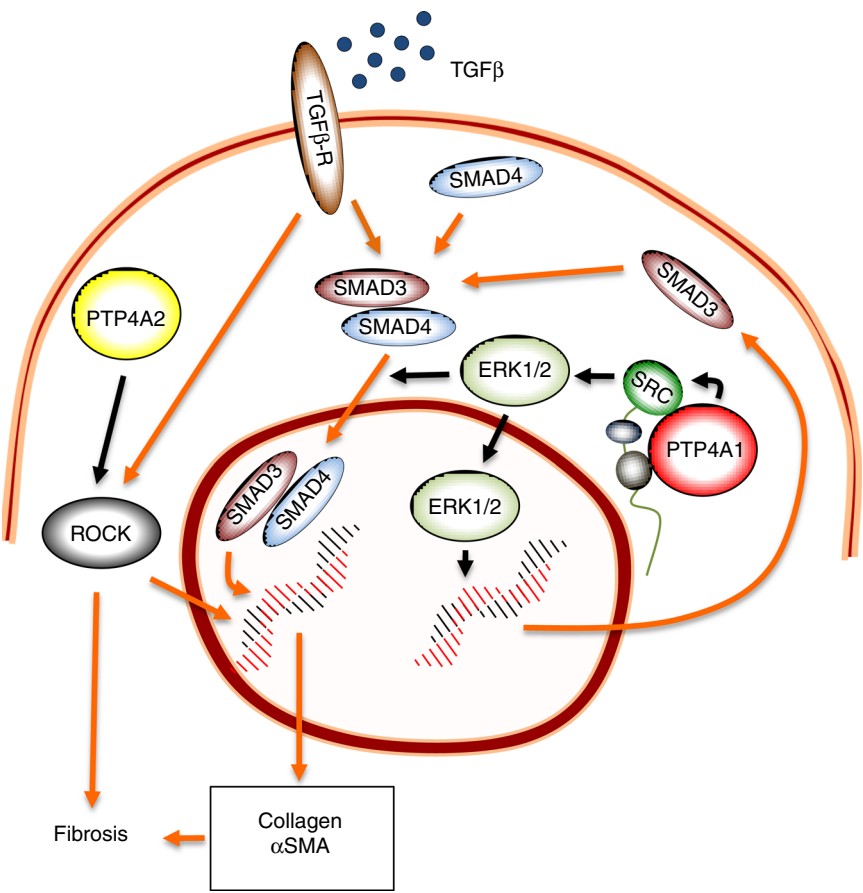

**Fig. 8** PTP4A1 promotes TGFβ signaling in DFs. PTP4A1 interacts with SRC and promotes SRC half-life, which in turn enhances activation of the MAPK pathway. Activated ERK1/2 induces transcription of SMAD3 and promotes TGFβ-driven SMAD3 nuclear translocation, which boosts pro-fibrotic TGFβ signaling. PTP4A2, despite its high identity with PTP4A1, does not interact with SRC or affect the MAPK–SMAD3 pathway. However, it also promotes pro-fibrotic signaling, perhaps through selective activation of the ROCK pathway. Orange continuous connectors: TGFβ signaling pathways. Black connectors: PTP4A1/PTP4A2 functions in human DFs

of SSc are characterized by enhanced immune-mediated inflammation, it is also possible that additional cytokines/inflammatory stimuli boost PTP4A1/PTP4A2 expression in early vs. late SSc biopsies.

Our transcriptomic study in resting NHDF lines subjected to silencing of PTP4A1 showed that PTP4A1 promotes in these cells the expression of a pro-fibrotic gene signature. Importantly, PTP4A1 appeared to critically increase the expression levels of SMAD3, a key mediator of TGFβ signaling, leading to the hypothesis that PTP4A1 plays a positive role in TGFβ signaling of NHDF lines. Subsequent experiments showed that PTP4A1 indeed enhances pro-fibrotic TGFβ signaling in NHDF lines as well as in lung fibroblasts and that deletion of PTP4A1 in vivo exerts a protective effect against bleomycin-induced fibrosis. PTP4A2 similarly promotes pro-fibrotic TGFβ signaling in NHDF lines and bleomycin-induced fibrosis in vivo. These experiments led us to define PTP4A1 and PTP4A2 as novel promoters of TGFβ signaling, a function that they currently share with very few additional tyrosine phosphatases[12, 13]. The bleomycin-induced dermal fibrosis model is widely used for modeling late stages of excessive accumulation of collagen and other extracellular matrix components in scleroderma[9, 27]. Thus, our data also points to a possible role for PTP4A1 and PTP4A2 in mediating fibrosis in SSc. The fibroblast-specific expression of the fibrotic markers that correlate with PTP4A1 expression in mice and the phenotype of the COL1A1-Cre conditional KO mouse strongly supports the idea that PTP4A1 promotes fibrosis

through an effect on fibroblasts. However, effects of PTP4A1 and/or PTP4A2 on additional cell types cannot be ruled out and warrant further investigation.

We next focused on elucidating the mechanism of action of PTP4A1 on TGFβ signaling in NHDF lines. Our data suggest a model where PTP4A1 enhances TGFβ signaling by driving basal expression of SMAD3 and promoting SMAD3 nuclear translocation/retention. Since we find that in NHDF lines PTP4A1 is a critical promoter of basal ERK phosphorylation and that ERK activity controls SMAD3 expression and TGFβ-induced nuclear translocation, we propose that PTP4A1 controls SMAD3 expression and cytosolic-nuclear shuttling through promotion of basal ERK activity. Surprisingly, in NHDF lines PTP4A2 does not influence SMAD3 expression or nuclear translocation and only minimally affects ERK activity. While this observation was unexpected based on the reports that both PTP4A1 and PTP4A2 promote ERK signaling in cancer cells, it does support the possibility that, in NHDF lines, only PTP4A1 controls TGFβ signaling through the ERK pathway. Another shared function of PTP4A1 and PTP4A2 in cancer cells is the promotion of Rho-GAP and downstream Rho activation through an action on p115 Rho[18, 32]. However, in NHDF lines promotion of ROCK activity appeared to be an exclusive function of PTP4A2, suggesting that PTP4A1 and PTP4A2 control different pro-fibrotic signaling pathways in NHDF lines and perhaps in vivo. The apparent lack of functional overlap between PTP4A1 and PTP4A2 in NHDF lines may be due to cell-specific differences in enzyme localization

and regulation. In addition, most of the previous studies have been carried out in growth factor-stimulated cells, where cross-activation of different pathways, for example, between the Rho and ERK pathway[34], might blur functional differences between the two phosphatase homologs.

An assessment of signaling upstream ERK led us to the observation that SRC activity is decreased in NHDF lines subjected to *PTP4A1* silencing. This finding correlated with decreased levels of SRC at the protein level, in *PTP4A1*-silenced NHDF lines, while SRC mRNA levels remained unaffected. Thus, we suggest a model where PTP4A1 promotes ERK signaling in NHDF lines through enhancement of basal SRC half-life. It has been reported that silencing of PTP4A1 in human lung cancer cells resulted in reduced SRC half-life, though the molecular mechanism behind the regulation of SRC by PTP4A1 was not addressed[39]. Since auto-phosphorylation of SRC on the activatory Y416 leads to Cul5-mediated ubiquitination and degradation[40, 41], one might hypothesize that PTP4A1 is an inhibitor of SRC auto-phosphorylation. To date, only one report has addressed the effect of PTP4A1 on SRC activity in HEK 293 cells, concluding that PTP4A1 promotes SRC Y416 phosphorylation[38]. In our experiments, consistent with the known model of regulation of SRC half-life, by Y416 phosphorylation, silencing of PTP4A1 resulted in increased phosphorylation of SRC on Y416 in resting NHDF lines. In these cells, we also observed increased phosphorylation of SRC on the inhibitory Y527. The latter phenomenon might be a compensatory mechanism, triggered by increased basal activation of SRC. Another scenario is that lack of PTP4A1 not only directly promotes auto-phosphorylation of SRC Y416, but also phosphorylation of SRC Y527. However, we could not find significant difference in expression levels of CSK—the kinase that phosphorylates SRC Y527—in NHDF treated with PTP4A1 ASO vs. Cntrl ASO (Supplementary Fig. 14A, B). Moreover, PTP4A1 was unable to in vitro dephosphorylate SRC on pY527 (Supplementary Fig. 14C, CD45 is a known SRC pY572 phosphatase and was used as a control in this experiment). Irrespective, our data suggest that PTP4A1 operates as a basal SRC rheostat, by inhibiting SRC degradation through decreased phosphorylation at Y416, and priming SRC for activation by decreased phosphorylation at Y527. Importantly, silencing of PTP4A2 did not influence SRC expression or stability, as well as the activation of MAPKK and MAPKKK upstream ERK in NHDF lines, further supporting the importance of SRC in the mechanism of action of PTP4A1 in these cells.

We finally sought to clarify how PTP4A1 regulates SRC phosphorylation on Y416. Our data suggest that inhibition of SRC phosphorylation by PTP4A1 does not depend on PTP4A1 catalytic activity. By comparing PTP4A1 and PTP4A2 in parallel assays, we show that SRC activation in *PTP4A1*-silenced NHDF lines correlates with a selective ability of PTP4A1 to physically interact with SRC and inhibit its auto-phosphorylation in in vitro kinase assays. Thus, we propose that PTP4A1 inhibits basal SRC auto-phosphorylation and degradation through a direct interaction with SRC. This model is consistent with the notion that PTP4A phosphatases display only weak activity in vitro on general phosphatase substrates[49] and agrees with reports showing that at least two of the major functions of PTP4A1 and/or PTP4A2—regulation of CNNM channels, and of RhoGAP function—are based on protein–protein interaction, rather than substrate dephosphorylation[18, 35, 50]. Among tyrosine phosphatases devoid of catalytic activity, the pseudophosphatase STYX can operate as an ERK anchor and inhibits ERK activity[51]. However, the molecular basis of ERK inhibition by STYX has not yet been elucidated. The observation that PTP4A1 inhibits SRC auto-phosphorylation in a catalytic activity-independent manner resemble the mechanism of regulation of two-component phosphohistidine-based systems by some prokaryotic bacterial phosphatases[52]. The molecular details of the specific interaction between SRC and PTP4A1 warrant further investigation. Possible mechanisms of SRC inhibition by the PTP4A1–SRC complex include steric inhibition of substrate accessibility and/or induction of kinase conformational changes. It also remains to be clarified whether the interaction of SRC with PTP4A1 influences the ability of SRC to phosphorylate its downstream substrates. The limited SRC-preservation activity of PTP4A1 in vitro could be sufficient to limit regulation of SRC activity and half-life by PTP4A1 to resting cells, thus avoiding significant limitations to SRC activation after growth factor or other SRC-dependent cell stimuli. Alternatively, specific mechanisms might be in place to remove PTP4A1 from SRC, after cells are subjected to specific stimuli.

In conclusion, we showed that PTP4A1, a tyrosine phosphatase overexpressed in DFs of patients with SSc, enhances canonical pro-fibrotic TGFβ signaling in these cells. We propose a model (Fig. 8) where, in resting fibroblasts, PTP4A1 directly interacts with SRC and inhibits its basal phosphorylation on Y416, thus limiting SRC degradation. The larger pool of available SRC, in combination with decreased phosphorylation of SRC on inhibitory Y527, promotes SMAD3 expression through the ERK pathway. Increased SMAD3 expression and perhaps SMAD3 phosphorylation by ERK in turn increase responsiveness of NHDF to TGFβ stimulation. Our results are consistent with previous observations that SRC activity is essential for TGFβ responsiveness of DFs[53].

By clarifying the relationship between SRC and PTP4A1 and showing that the functional overlap between PTP4A1 and PTP4A2 is less extensive than expected, our findings significantly contributes to clarification of the biology of the elusive PTP4A subclass of PTPs. Our study also suggests that interfering with the interaction between PTP4A1 and SRC might represent a therapeutic strategy in fibrotic diseases where TGFβ is believed to play a pathogenic role.

## Methods
**Antibodies and other reagents**. The mouse anti-PTP4A1/2 (clone 42) antibody was purchased from EMD Millipore (Billerica, MA). The rabbit anti-SMAD3 (catalog number 9513), rabbit anti-pSMAD3 (S423/S425, clone C25A9), rabbit anti-pERK1/2 (T202/Y204, catalog number 9101), rabbit anti-HA (clone C29F4), rabbit anti-MEK1/2 (catalog number 9122), rabbit anti-pMEK1/2 (S217/221, catalog number 9121), rabbit anti-pRAF (S259, catalog number 9421), rabbit anti-pPLCγ (Y783, catalog number 2821), rabbit anti-SRC (catalog number 2108), mouse anti-SRC (clone L4A1), rabbit anti-pSRC (Y416, catalog number 2101), rabbit anti-pSRC (Y527, catalog number 2105), and rabbit anti-GAPDH (clone 14C10) antibodies were purchased from Cell Signaling Technology (Danvers, MA). The mouse anti-αSMA (clone 1A4), Cy3-conjugated mouse anti-αSMA (clone 1A4), and mouse anti-αTubulin (clone DM1A) antibodies were purchased from Sigma-Aldrich (St. Louis, MO). The rabbit anti-αSMA antibody (catalog number 5694) was purchased from Abcam (Cambridge, MA). Secondary antibodies for western blotting and IF were purchased from GE Healthcare Life Sciences (Buckinghamshire, UK) and eBioscience (San Diego, CA). Antisense oligonucleotides were from Gene Tools LLC (Philomath, OR). Transforming growth factor beta 1 (TGFβ1) was purchased from eBioscience (San Diego, CA). Bleomycin sulfate, the SB505124 TGFβ inhibitor, the SU6656 SRC inhibitor, and the SCH772984 ERK inhibitor were purchased from Selleck Chemicals (Houston, TX). Recombinant GST-tagged SRC (rSRC), recombinant CD45 (rCD45), recombinant HA-tagged PTP4A1 (rPTP4A1), or PTP4A2 (rPTP4A2) were purchased from R&D Systems (Minneapolis, MN). Recombinant GST-tagged CSK (rCSK) was purchased from EMD Millipore. Unless specified, chemicals and all other reagents were purchased from Sigma-Aldrich.

**SSc and healthy skin samples**. SSc and healthy skin biopsies and DFs lines were obtained from the Thomas Jefferson University (TJU IRB #06F.186), the UC San Diego Division of Rheumatology (UC San Diego IRB #140485), the UCSF Scleroderma Center (UCSF IRB #15-16463), and the ASL Avezzano Sulmona L'Aquila (protocol number #0214409/16). Healthy skin samples were also obtained through the NDRI tissue bank (Philadelphia, PA). Lung fibroblast cell lines were obtained from ATCC (Manassas, VA) and Lonza (Basel, Switzerland). All patients met the

2013 American College of Rheumatology/European League against Rheumatism criteria for SSc[1], and fulfilled the criteria for dcSSc or lcSSc. dcSSc patients recruited < 3 years from first non-Raynaud manifestation were classified as early dcSSc and patients recruited > 3 years from first non-Raynaud manifestation were classified as late dcSSc[15]. The cohorts included 40 male and female Caucasian and African American patients with SSc, or 12 female Caucasian healthy subjects, aged between 18 and 80 years old (see Supplementary Table 1 for more details about the SSc cohorts). All patients and controls signed a consent form approved by the local institutional review boards (TJU IRB #06F.186, UC San Diego IRB #140485, UCSF IRB #15-16463, and ASL Avezzano Sulmona L'Aquila protocol number #0214409/16).

DFs were cultured in complete Dulbecco's Modified Eagle's Medium from Thermo Fisher Scientific (Waltham, MA) with 10% fetal bovine serum (FBS) from Omega Scientific (Tarzana, CA), 100 units/ml of penicillin and 100 µg/ml streptomycin from Thermo Fisher Scientific at 37 °C in a humidified 5% $CO_2$ atmosphere. For all experiments, fibroblasts were used between passages 3 and 8, and cells were synchronized in serum-starved media with 0.1% FBS for 24 h prior to analysis or functional assays. None of the cell lines used were listed as a misidentified cell line in the International Cell Line Authentication Committee database. The absence of mycoplasma contamination was confirmed using the MycoAlert Mycoplasma Detection Kit from Lonza.

**IF analysis of human skin.** Paraffin-embedded sections of human skin were deparaffinized, rehydrated, and treated for 10 min with boiling citrate antigen-retrieval buffer (1.9 mM citric acid, 10 mM Tris-sodium citrate, pH 6.0). Slides were blocked with 5% goat serum for 1 h at room temperature (RT) and incubated overnight at 4 °C with mouse anti-PTP4A1/2 antibody and rabbit anti-αSMA (1:200 dilution in 5% bovine serum albumin (BSA)). Slides were washed and incubated for 1 h with Alexa Fluor 488-conjugated goat anti-mouse IgG Alexa Fluor 568-conjugated goat anti-rabbit IgG (1:500 dilution in 5% BSA). Nuclei were stained for 10 min at RT with 5 µg/ml Hoechst 33242 from Thermo Fisher Scientific. Images were obtained using a Fluoview FV10i confocal microscope or a BX53 fluorescence microscope from Olympus (Tokyo, Japan). PTP4A1/2 fluorescence densitometry was assessed only in cells co-stained with αSMA in three different fields for each slide. ImageJ software was used for signal quantification.

**Mice.** Animal experiments were conducted in accordance with Institutional Animal Care and Use Committee-approved protocols at Purdue University (#1511001324) and at the La Jolla Institute for Allergy & Immunology (#AP140-NB4). Eight-week-old female C57BL/6 mice were obtained from the Jackson Laboratory (Bar Harbor, ME). C57BL/6 mice carrying deletion of *PTP4A2* (PTP4A2 KO) have already been described[26]. A heterozygous gene-trapped embryonic stem (ES) cell line (Cell No.: CC0606; Mouse strain: 129P2/OlaHsd) containing an insertional mutation in the mouse *PTP4A1* locus was purchased from the Mutant Mouse Regional Resource Centers (MMRRC). ES cells were injected into blastocysts from C57BL/6J mice at the Transgenic and Knock-Out Mouse Core of the Indiana University Simon Cancer Center. Heterozygous males resulting from successful germ-line transmission of the mutant allele were inter-crossed with C57BL/6J females to generate F1 offspring. The PTP4A1-targeted ES cell line was subjected to genetic confirmation assessment at MMRRC, which included long-range PCR to confirm proper targeting, vector copy number count, and Y chromosome verification. Reverse transcription polymerase chain reaction (RT-PCR) assessment showed that PTP4A1 KO mice do not express PTP4A1. In addition, RT-PCR revealed that beta-geo is not expressed as well in 12-week-old female PTP4A1 KO mice. 8–12-week-old female PTP4A1 and PTP4A2 KO mice used in this study were on C57BL/6:129P2 mixed genetic background. LacZ primers for beta-geo expression, 5′-GATCGCGTCACACTACGTCT-3′ (forward) and 5′-GTGGCCTGATTCATTCCCCA-3′ (reverse), were purchased from IDT. PTP4A1^fl/fl homozygous mice on C57BL/6 background have been already described[10] and were mated with C57BL/6.Cg-Tg(COL1A1-Cre/ERT2)1Crm/J mice from the Jackson Laboratory. Cre expression was induced postnatal in 7-week-old female PTP4A1^fl/fl—COL1A1 Cre mice through intraperitoneal injections of 2 mg tamoxifen dissolved in 100 µl corn oil for 5 consecutive days.

**Bleomycin-induced fibrosis model.** Experimental dermal fibrosis was induced in 8–12-week-old female mice through seven subcutaneous injections every other day of 50 µg bleomycin sulfate dissolved in 100 µl phosphate-buffered saline (PBS). In the PTP4A1^fl/fl—COL1A1 Cre mice, bleomycin was injected 3 days after the last tamoxifen injection. All treatment groups consisted of at least five mice to detect a significant effect. We focused on female mice exclusively, since SSc overwhelmingly affects women vs. men. The severity of fibrosis was assessed at the bleomycin injection sites using an AxioScan Z1 slide scanner (Zeiss, Jena, Germany) on trichrome-stained skin and lung sections, prepared at the La Jolla Institute histopathology core. The distance between the epidermal–dermal junction and the dermal–subcutaneous fat junction was calculated in triplicate in three consecutive skin sections from each animal. Additional fragments of affected skin were used to assess collagen deposition using the Sircol assay kit from Biocolors (Belfast, UK) and *ACTA2* mRNA expression. No randomization or blinding were used.

**Quantitative real-time polymerase chain reaction.** RNA was extracted using mini and micro RNeasy kits from Qiagen (Hilden, Germany). Complementary DNA (cDNA) was synthesized using the SuperScript III First-Strand Synthesis System from Thermo Fisher Scientific. qPCR was performed using a LightCycler 480 from Roche (Basel, Switzerland), with individual primer assays and SYBR Green qPCR Master Mix from Qiagen. Efficiency of the primer assays was guaranteed by the manufacturer to be > 90%. Each reaction was analyzed in triplicate and data were normalized to the expression levels of the housekeeping genes *POLR2A* or *GAPDH*. Absence of genomic DNA contamination was confirmed using control reactions lacking the reverse transcriptase enzyme during the cDNA synthesis step.

**Western blotting.** Cells were lysed in ice-cold RIPA buffer (50 mM TrisHCl [pH 8.0], 150 mM NaCl, 5 mM EDTA, 1% NP40, 0.5% sodium deoxycholate, 0.1% sodium dodecyl sulfate) or TNE buffer (50 mM TrisHCl [pH 8.0], 150 mM NaCl, 5 mM EDTA) with protease and phosphatase inhibitor cocktails from Roche. Protein concentrations in cell lysates were determined using a bicinchoninic acid protein assay kit from Thermo Fisher Scientific. Proteins were loaded on 10% or 4–20% Tris-Glycine gels from Thermo Fisher Scientific, electrotransferred onto nitro-cellulose membranes from GE Healthcare Life Sciences (Pittsburgh, PA) and incubated overnight at 4 °C with specific antibodies (1:1000 dilution in 5% BSA). Immunoreactivity was revealed by incubation with host-specific secondary antibodies conjugated with horseradish peroxidase (HRP), followed by detection with Luminata Crescendo Western HRP substrate from EDM Millipore. Images were acquired using a G:BOX chemi system from Syngene (Frederick, MD) and protein bands analyzed with the ImageJ software. Protein expression was normalized to the expression levels of tubulin, GAPDH, or non-phosphorylated corresponding proteins. Uncropped images of the blots are provided in Supplementary Information.

**Treatment with cell-permeable antisense oligonucleotide.** NHDF and NHLF fibroblasts were treated for 7 days with 2.5 mM control or targeting ASOs.

PTP4A1 and PTP4A2 ASOs were designed against the pre-mRNA splicing junction between intron 2 and exon 3. PTP4A1 ASO2 was designed against the pre-mRNA splicing junction between exon 3 and intron 3. PTP4A1 ASO3 was designed against the pre-mRNA splicing junction between intron 3 and exon 4. ASOs were replaced in fresh culture medium after 3 days and in serum-starvation medium after 6 days. RT-PCR was performed on ASO-treated fibroblasts cDNA with PTP4A1-specific (forward: 5′-ACAATCCAACCAATGCGACC-3′; reverse: 5′-GCTGTTAAAAGCTCCACGCC-3′) and PTP4A2-specific (forward: 5′-GGAGTGACGACTTTGGTTCG-3′; reverse: 5′-TCAAAGCAAGTGCAACCAGC-3′) oligos from IDT (San Diego, CA) using a S1000 thermal cycler from Bio-Rad (Irvine, CA).

**NGS on NHDF.** Three NHDF lines were plated in triplicate, incubated with control or PTP4A1 ASOs and serum-starved for 24 h prior the experiment. RNA was extracted using micro RNeasy kits from Qiagen and 500 ng of total RNA for each condition were used to generate mRNA-focused libraries on a Biomek FXP automation platform from Beckman Coulter (Carlsbad, CA) with the TruSeq-Stranded mRNA HT Sample Prep Kit from Illumina (San Diego, CA). NGS was performed at the La Jolla Institute sequencing facility with a Hiseq 2500 from Illumina.

**TGFβ stimulation of fibroblasts.** NHDF or NHLF lines were incubated with control or targeting ASOs, serum-starved for 24 h and stimulated with 20 ng/ml human TGFβ1 or vehicle as control. Cells were lysed for qPCR or western blotting 24 h or 15 min after TGFβ1 stimulation, respectively.

**ERK inhibitor treatment.** NHDF lines were serum-starved for 24 h and treated with 50 µM SCH772984 ERK inhibitor or DMSO as control. Cells were lysed for *SMAD3* qPCR 24 h after the treatment.

**SMAD3 localization.** NHDF lines were incubated with control ASO, PTP4A1 ASO, PTP4A2 ASO or DMSO, SU6656 SRC, and SCH772984 ERK inhibitors, serum-starved for 24 h and stimulated with 20 ng/ml human TGFβ1 or vehicle as control. Thirty minutes after stimulation, cells were fixed in 4% formaldehyde at RT and permeabilized in PBS with 1% BSA and 0.4% Triton X-100 for 20 min at RT. Cells were then incubated overnight at 4 °C with rabbit anti-SMAD3 antibody (1:200 dilution in PBS with 1% BSA and 0.1% Triton X-100), washed and incubated for 1 h with Alexa Fluor 568-conjugated goat anti-rabbit IgG (1:500 dilution in PBS with 1% BSA and 0.1% Triton X-100). Nuclei were stained for 10 min at RT with 5 µg/ml Hoechst 33242 from Thermo Fisher Scientific. Sample images were obtained using a Fluoview FV10i confocal microscope from Olympus. For SMAD3 fluorescence densitometry three different fields were assessed for each NHDF line. ImageJ software was used for signal quantification.

**CHIP qPCR.** NHDF lines were incubated with control or PTP4A1 ASO, serum-starved for 24 h and stimulated with 20 ng/ml human TGFβ1 or vehicle as control. Then cells were fixed in 1% formaldehyde for 15 min at RT. After sonication,

chromatin was immunoprecipitated with the ChIP assay kit from Millipore using a rabbit anti-pSMAD3 (S423/425) antibody overnight at 4 °C. The eluted DNA was purified with ChIP DNA Clean & Concentrator from Zymo Research (Irvine, CA) and used for COL1A2 promoter qPCR. 10% input for each condition was used for normalization. Human COL1A2 promoter primers[54], 5′-TCTGCCCATGTCGGG GCT-3′ (forward) and 5′-TGCCTCCAAAAGGGCCTCC-3′ (reverse), were purchased from IDT.

**Constructs, transfections, and luciferase reporter assays**. WT and mutant PTP4A1 and PTP4A2 human genes cloned in pCDNA4 have been already described[30]. ΔPTP4A1, lacking the exon 4 and cloned in the pET28 vector, was obtained from Genscript (Piscataway, NJ). The expression of 6× histidine-tagged ΔPTP4A1 was induced for 4 h with 1 mM isopropyl β-D-1-thiogalactopyranoside in E. coli BL21 (DE3) strain and purified with an IMAC nickel resin and a NGC Chromatography Systems from Bio-Rad (Hercules, CA). The Cignal SMAD reporter kit was purchased from Qiagen. Transfections were conducted in HEK 293T cells from ATCC using Lipofectamine 3000 from Thermo Scientific. Cells were stimulated with 20 ng/ml human TGFβ1 24 h after transfection. Luciferase activity was assessed using the Dual-Luciferase Reporter Assay System from Promega (Madison, WI). Renilla luciferase activity was used to normalize firefly luciferase activity.

**Enzymatic and co-precipitation assays**. NHDF lines were incubated with control or targeting ASOs, serum-starved for 24 h and stimulated with 20 ng/ml human TGFβ1 or vehicle as control. Cells were lysed 15 min after stimulation in ice-cold TNE buffer and ROCK activity was measured with the Rho-associated kinase activity assay from EMD Millipore. SRC dephosphorylation was assessed in the presence of recombinant PTP4A1 or PTP4A2. Briefly, 50 ng of GST-tagged rSRC was incubated with glutathione sepharose high-performance medium from GE Healthcare Life Sciences and activated in kinase assay buffer (60 mM HEPES pH 7.5, 5 mM MgCl₂, 5 mM MnCl₂, 1 mM DTT, 1 mM Na₃VO₄, 1 mM ATP) for 10 min at 30 °C in agitation. The rSRC-medium reaction complex was washed and incubated with 1.5 μM rPTP4A1, rPTP4A2, or BSA as control in 60 mM HEPES pH 7.5, 5 mM MgCl₂, 5 mM MnCl₂, 1 mM DTT for 2 h at 37 °C in agitation. The beads were then either assessed for rSRC phosphorylation on Y416 or washed for assessing whether rPTP4A1, ΔPTP4A1, or rPTP4A2 were bound to rSRC. For kinase assays in the presence of PTP4A1 and PTP4A2, 50 ng of rSRC was incubated in kinase assay buffer (60 mM HEPES pH 7.5, 5 mM MgCl₂, 5 mM MnCl₂, 1 mM DTT) together with 1.5 μM rPTP4A1, rPTP4A2, or BSA as control. rSRC was then assessed for its phosphorylation on Y416 as described above. For CSK kinase assay, 20 ng of rSRC was incubated at 30 °C for 2 h in kinase assay buffer (60 mM HEPES pH 7.5, 5 mM MgCl₂, 5 mM MnCl₂, 1 mM DTT, 1 mM Na₃VO₄, 1 mM ATP) together with 50 ng rCSK, 100 ng rPTP4A1, and 200 ng rCD45. rSRC was then assessed for its phosphorylation on Y527 as described above.

**PTP4A1 enzymatic activity**. 40 μg/ml of BSA, wtPTP4A1, or ΔPTP4A1 were incubated at 37 °C for 8 h with 5 mM p-nitrophenyl phosphate in 50 mM HEPES, 10 mM DTT, pH 7.5. Absorbance at 410 nm was recorded with an Infinite M1000 (Tecan, San Jose, CA) and specific activity was reported as pmol/min/μg.

**Immunoprecipitation**. HA-tagged PTP4A1 WT and C104S cloned in pCDNA4 vector[30] or HA-tagged PTP4A2 WT and C101S cloned in pCDNA4 vector (obtained from Mutagenex [Suwanee, GA]) were transfected in HEK 293T cells (ATCC) using Lipofectamine 3000 from Thermo Scientific. 48 h after transfection, cells were lysed in ice-cold TNE buffer with a protease inhibitor cocktail from Roche and 10 mM Na₃VO₄, then incubated for 2 h at 4 °C with rabbit anti-SRC or anti-HA antibody 1:100 and immunoprecipitated 2 h at 4 °C on protein G Sepharose 4 fast flow from GE Healthcare Life Sciences. The co-precipitation of HA-tagged PTP4A1 or PTP4A2 with SRC was assessed by western blotting.

**Statistical analysis**. Sample sizes were selected based on our experience with the above-mentioned assays in order to achieve sufficient power to detect biologically relevant differences in the experiments being conducted with an α error (two-tailed) <0.05. The two-tailed Mann–Whitney test or the two-tailed Welch's t-test were performed where appropriate as reported in the figure legends. For experiments where variance was not similar between the groups being compared, the two-tailed Wilcoxon test or the two-tailed paired t-test were used. A comparison was considered significant if p was less than 0.05. All statistical analyses were performed using GraphPad Prism software. Raw data sets for the figures in main text are provided in Supplementary Data 1. Raw data sets for the figures in Supplementary Information are provided in Supplementary Data 2.

NGS differential expression was analyzed with DeSeq software and genes with threshold expression higher than 100, log₂ fold change expression higher than 0.5 and adjusted p-values less than 0.01 were considered significantly affected by PTP4A1 silencing in NHDF lines. NGS molecular functions, biological processes, and cellular compartments analysis were performed with GO-Elite software.

**Data availability**. Sequence data that support the findings of this study have been deposited in GEO with the primary accession code GSE102864. The authors declare that all other data supporting the findings of this study are available within the article and its Supplementary Information files.

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

## Acknowledgements

This work was supported in part from the National Institute of Arthritis and Musculoskeletal and Skin Diseases (grant AR069822) to N.B. and from the National Cancer Institute (grant CA069202) to Z.Y.Z. This is manuscript #3049 from the La Jolla Institute for Allergy and Immunology.

## Author contributions

C.S., S.M.S., F.B., S.A.J., R.G., Z.Y.Z. and N.B. conceived and designed the study. C.S., Y.B., S.M.S., P.D.B., P.C., E.S., S.P.V., V.C., A.C. and W.B.K. acquired the data. C.S., S.M.S., R.G. and N.B. analyzed and interpreted the data. K.H.K. provided an essential reagent. All authors were involved in drafting the article or revising it critically for important intellectual content, and all authors approved the final version to be published.

## Additional information

**Competing interests:** This work was supported in part by investigator-initiated grants from Kyowa Kirin Pharmaceutical Research Inc. (formerly Kyowa Hakko Kirin California Inc.) to N.B. and from Illumina Inc. to C.S. The La Jolla Institute for Allergy and Immunology holds a pending patent application, "PTP4A1 as a therapeutic target for fibrotic diseases and disorders including systemic sclerosis", with N.B., C.S. and S.M.S. named as inventors. The remaining authors declare no competing financial interests.

