## [Peer review file · Nature Communications]

Reviewers' comments:

Reviewer #1 (Remarks to the Author):

The authors have used a variety of technical approaches to explore the potential relevance of the PTP4A1 and PTP4A2 phosphatases the pathogenesis of the human fibrotic diseases scleroderma (SSc). The starting point is profiling of phosphatases that identified PTP4A1 phosphatase as an upregulated candidate. This was undertaken in a small number of SSc fibroblast strains and later confirmed in others. To delineate the relevance of this to TGFbeta signalling and fibrosis other models were explored and a fibroblast specific mouse deletion was engineered. Knock down experiments in human fibroblasts were also performed. This study is comprehensive and uses a convincing set methods but essentially is confirmation of the perturbation of TGFbeta regulated pathways in systemic sclerosis and their relevance to fibrosis. It seems unlikely that this is a pivotal or unifying mechanism that is likely to be disease specific to systemic sclerosis. This reduces my enthusiasm for the study but technically and in concept it is of high quality.

Specific comments

1. The heterogeneity of SSc and potential effects of treatments needs to be accounted for and so it would be interesting to explore this is a larger number of well characterized human SSc samples including limited as well as diffuse subset.
2. The use of TGFbeta blocking strategies in SSc is not as clearly beneficial from clinical trial data in human SSc as suggested by the authors and the discussion and top line comments about this could be modified. Effects in one study negative and in a more recent one only modest.
3. It would be helpful to explore the mutant mouse phenotype in more detail and to discuss in more detail any differences in the phenotype of conventional null mice. The authors should confirm if the tamoxifen-dependent Cre-ER mice used for conditional deletion is driven by Col1a1 or Col1a2 – the latter is much more fibroblast specific the former drives expression in may type 1 collagen producing cells.
4. It would be helpful to test at least one fibrotic model in the fibroblast deleted mouse strain to assess prevention or treatment benefit. This could be explored by postnatal genetic recombination.
5. Other TGFbeta regulated transcripts might have been expected to change in fibroblasts and the effect of TGFbeta stimulation should be explored. COL1A1 and COLA2. Exploration at a protein level as these proteins not always transcriptionally regulated. CTGF is notably absent in the gene list described in the manuscript.

Overall, although this is very elegant work it is not clear how the findings are distinguished from those of many other studies that confirm activation and dysregulation of canonical and non-canonical TGFbeta signalling in SSc fibroblasts. This should be clarified in the abstract and conclusions of the paper.

Reviewer #2 (Remarks to the Author):

The present manuscript entitled "PTP4A1 phosphatase is overexpressed in systemic sclerosis fibroblasts and promotes TGFβ signaling" shows the implication of PTP4A1 (and PTP4A2) as molecular regulators of TGFβ signaling in healthy dermal fibroblasts and connects this, for PTP4A1, to the pathological context of systemic sclerosis.

Given the fact that protein tyrosine kinases are considered as potential therapeutic targets against SSc, this manuscript addresses the interesting question whether the molecular processes involved in SSc could also be modulated by protein tyrosine phosphatases. Among all PTPs, the authors have found PTP4A1 overexpressed in patient samples compared to healthy samples. PTP4A2 is also highly expressed, but not significantly elevated in patient samples. Therefore, the authors focus on PTP4A1 in the following study. These initial findings and the subsequent results showed in

the manuscript are very relevant, adding novelty and contributing to a better understanding of the PTP4A group of phosphatases, which have been extensively described overexpressed in different types of cancer.

The novelties of this study include the following:

- 1) The involvement of PTP4A1 in TGFbeta signaling: This is robustly described, and is supported by previously published data such as promotion of SRC activity and MAPK signaling.
- 2) The differences regarding the involvement of PTP4A2 compared to PTP4A1 in this signaling process: These are largely clear.
- 3) The (potential) involvement in the disease. This is not so well established.
- 4) To me, the key impact of this work is in the direct interaction of PTP4A1 with SRC and its consequences, because many other indirect involvements in signaling pathways have been described previously (such as in MAPK and SRC signaling), leaving the question of how the phosphatase promotes signaling mechanistically. Because of the importance for the impact of the paper, the data should be completely convincing, which is currently not the case.

I therefore think the manuscript must be strengthened in order to be published in Nature Communications.

Major comments:

- 1) The authors mix up a healthy versus a disease background throughout the paper. They find PTP4A1 overexpressed in dcSScDF cells compared to NHDF cells. All molecular biology experiments on the mechanism are done in NHDF cells. They connect to the disease state again with their knock out mouse models by applying a bleomycin-induced dermal fibrosis test and note that the knock out mice are protected from skin fibrosis. However, no mechanistic studies are done in the mouse. Then in the discussion (page 22, line 18,19) they state, "...we show that PTP4A1, a tyrosine phosphatase overexpressed in dermal fibroblasts of SSc patients, enhances canonical pro-fibrotic TGFbeta signaling in these cells." This is not correct as they show that it is involved in canonical pro-fibrotic TGFbeta signaling in NHDF cells from healthy humans. While it is very likely that PTP4A1 acts in SSc through this mechanism, it is not shown, and it could also act for example by taking over PTP4A2 actions or other processes dependent on the higher abundance of PTP4A1 in the cells. The authors need at least to check SRC levels (protein and phosphorylation on the two sites) in SSc cells in order to connect their studies in NHDF cells to the disease.
- 2) In order to really connect PTP4A1-mediated stabilization of SRC to the Smad3 phenotype and MAPK signaling downstream, they should also use a SRC inhibitor in NHDF (and SSc) cells (for Fig. 7), like they did with Smad3 localization and ERK inhibition (Fig. S7 and S6). The inhibition of SRC should result in a similar phenotype and molecular profile as PTP4A1 knock down. This direct relation would strongly corroborate their working model.
- 3) In figure 7, the influence of the catalytic activity of PTP4A1 in the TGFβ signaling is addressed. The luciferase assay including the catalytic mutant (7A) shows no effect of the PTP4A1 catalytic activity, when overexpressed, on the SMAD activity. To further demonstrate this, knockdown experiments are performed using an ASO (ASO3) targeting the exon where the crucial Arginine residue (Arg110) for the PTP4A1 catalysis is located. This is a rather unusual approach. Detailed information about how this approach works is missing in the text. My understanding of this approach (considering the information given in the supplementary figure S9) is that the ASO3 generates a truncated protein that lacks the exon 4 sequence including the Arg110, and the resulting protein is then presumably inactive. If this were correct, I would have the concern of the stability of the protein generated upon ASO3 treatment, since creating an internal deletion could lead to a misfolded and less stable protein. If that were the case, the results shown in Figure 7B, 7C and 7D where the knockdown with ASO3 does not give any effect and behaves similar to the control ASO, could then be also explained by a putative unstable/misfolded/still catalytically active protein mimicking wt PTP4A1 behavior rather than by a truncated protein only lacking catalytic activity. In order to confirm this, the presence of the truncated protein has to be shown (by MS or Western blot), the stability and folding (for example by CD and Tm measurement) of the resulting truncated protein should be studied and compared to the wt PTP4A1. It should also be made sure that the truncated protein does not have phosphatase activity, and that it still binds to SRC, as this is the proposed mechanism. Alternatively, an inhibitor for the PTP4A family could be used. Since the authors can distinguish PTP4A1 and 2 by the Smad3 read out, they can use a promiscuous PTP4A

inhibitor like thienopyridone to test for inhibited PTP4A1 action.

4) Figure 6E shows that the knockdown of PTP4A1 increases the phosphorylation levels of SRC at Tyr416 and Tyr527, indicating a regulation of SRC activation mediated by PTP4A1. The authors further investigate whether Tyr416 is indeed dephosphorylated by PTP4A1, detecting no dephosphorylation (fig. 7E). However, they do not show a similar experiment to see if SRC Tyr527 is a substrate of PTP4A1 in NHDF lines. Since the dephosphorylation of Tyr527 leads to the activation of SRC, it is worth to see if this residue is a substrate of PTP4A1 because this would be also one explanation of how SRC is kept active by PTP4A1. The authors detect the physical interaction between SRC and PTP4A1 and suggest the hypothesis of this interaction as a regulatory mechanism of SRC activation (preventing SRC basal autophosphorylation that would lead to an increase in SRC protein degradation), but it should be taken into account that a putative dephosphorylation of SRC Tyr527 by PTP4A1 could influence the state of SRC activation as well. This is necessary to really "clarify the relationship between SRC and PTP4A1" (page 23 line 4).

5) Figure S10: It is standard when showing a new interaction to do a reciprocal IP. The interaction has only been shown through Src IP, but the HA-PTP4A1 should be IPed as well. If the interaction is not affected by phosphatase activity (as they imply through using the C/S mutant in cells), why do they use the C/S mutants in the IP? Since PTP4A2 also binds to Src, do the authors think that there is an overlap here with their mechanisms after all? Or is it background? The respective reciprocal IP could also clarify this.

Minor comments:

1) Figures 5, 6 and 7 show in some cases the quantification of the phosphorylation levels (normalized to the corresponding total protein amount) of different proteins. However, the label in the y-axes refer to Relative expression and this term is not accurate in this case. I suggest to change the axis labels to "relative phosphorylation levels", for example.

2) Unlike PTP4A1, PTP4A2 does not regulate the expression levels of SMAD3 (fig. 3A), does not affect the SMAD3 nuclear translocation (fig. S7B) and does not promote ERK1/2 activation (fig. 5D). As pointed out by the authors, these observations indicate different pathways between PTP4A1 and PTP4A2. Nevertheless, although small, PTP4A2 enhances (1.6 fold) the luciferase activity (fig. 5E). I would like to know the authors opinion about how PTP4A2 could enhance the activity of SMAD3 (see also the detected interaction between PTP4A2 and SRC, mentioned above).

3) Discussion, page 18, line 17: typo: correct TFG by TGF

4) Discussion, page 22, line 12: "Src-inhibitory activity of PTP4A1" is a bit confusing, since in cells the overall outcome is more presence and activity of Src. It would be good to rephrase this somehow.

5) Two previous studies showed that PTP4A3 (the third member of the PTP4A phosphatases, not expressed in SScDF) exerts translational control of CSK, resulting in SRC activation (Liang et al., J Biol Chem. 2007 Feb 23;282(8):5413-9 and J Biol Chem. 2008 Apr 18;283(16):10339-46). Could PTP4A1 regulate CSK in the same way turning into SRC activation? This fact should be mentioned in the discussion part about SRC regulation. It would be interesting to reflect that these two PTP4A phosphatases could potentially have different or similar ways of regulate SRC (see also potential dephosphorylation of Y527).

6) Introduction: page 4 line 10: There is evidence that within the PTP4A family there are differences between them, such as promotion of EMT by PTP4A3, their different expression profiles (related to cell-type dependent actions as also mentioned in this manuscript), and their reported different in vitro substrate specificities. This sentence should be changed, as it is too general.

7) Page 7 lines 1-12: What is the difference between the early and late biopsies? What could be the reason that PTP4A1 overexpression is not evident at later stages?

8) For all knock-downs, the Western blots should be shown.

In conclusion, this work has the novelty, insight and impact to be considered for publication in Nature Communications, but these concerns should be addressed thoroughly beforehand.

NCOMMs-16-12525A-Z Response to Referees – revised text in the manuscript body is highlighted in yellow

Reviewer #1 (Remarks to the Author):

The authors have used a variety of technical approaches to explore the potential relevance of the PTP4A1 and PTP4A2 phosphatases the pathogenesis of the human fibrotic diseases scleroderma (SSc). The starting point is profiling of phosphatases that identified PTP4A1 phosphatase as an upregulated candidate. This was undertaken in a small number of SSc fibroblast strains and later confirmed in others. To delineate the relevance of this to TGFbeta signalling and fibrosis other models were explored and a fibroblast specific mouse deletion was engineered. Knock down experiments in human fibroblasts were also performed. This study is comprehensive and uses a convincing set methods but essentially is confirmation of the perturbation of TGFbeta regulated pathways in systemic sclerosis and their relevance to fibrosis. It seems unlikely that this is a pivotal or unifying mechanism that is likely to be disease specific to systemic sclerosis. This reduces my enthusiasm for the study but technically and in concept it is of high quality.

Specific comments

1. The heterogeneity of SSc and potential effects of treatments needs to be accounted for and so it would be interesting to explore this is a larger number of well characterized human SSc samples including limited as well as diffuse subset.

This is a reasonable comment. We have included a new subset of SSc limited patients to this revised version of the manuscript (see **new Fig. 1C-D** and **new table S1**). The new limited SSc are technically all late stage samples: because of the slow progression of disease in this variant of SSc, we could not find biopsies from any patient at early stage (<3y from onset of first non-Raynaud symptom). This brings the total number of patients that we collected cell lines or biopsies to 40, which we believe is a respectable number considering the frequency of SSc. Importantly, within our study, we made the same observation in patients from two independent populations (U.S. and Italy). We agree with the Reviewer that the exploration should be extended to an additional highly controlled set of samples. Indeed we are currently applying for funding to extend the study to additional sets of patients. Since, as the Reviewer points out, the mechanism of action of PTP4A1 is unlikely to be specific for SSc, we might focus our next investigations on more common fibrotic diseases, including IPF and liver fibrosis (see also our response to this Reviewer's closing remarks). We hope the reviewer will appreciate our effort to collect more samples and in consideration of the many additional significant, robust and novel observations reported in the manuscript, perhaps be a little forgiving on this point.

2. The use of TGFbeta blocking strategies in SSc is not as clearly beneficial from clinical trial data in human SSc as suggested by the authors and the discussion and top line comments about this could be modified. Effects in one study negative and in a more recent one only modest.

This is a very good point. Following this Reviewer's comment, we have modified the text about the use of TGFbeta blocking strategies in SSc to avoid overstatements.

3. It would be helpful to explore the mutant mouse phenotype in more detail and to discuss in more detail any differences in the phenotype of conventional null mice. The authors should confirm if the tamoxifen-dependent Cre-ER mice used for conditional deletion is driven by Col1a1 or Col1a2 – the latter is much more fibroblast specific the former drives expression in many type 1 collagen producing cells.

The tamoxifen-dependent Cre-ER driver used for conditional deletion is driven by the Col1a1 promoter, and this is now described in Fig. 4 legend and in the Material and Methods section. We apologize for not having made this more clear in the previous version. We agree with the Reviewer that Col1a1 drives expression in a wider range of collagen-expressing cells than Col1a2 (in particular Col1a1 also drives deletion in bone compared to Col1a2), however Fig. 4F suggests that at the skin level, our Cre driver led to a quite specific deletion in dermal fibroblasts.

4. It would be helpful to test at least one fibrotic model in the fibroblast deleted mouse strain to assess prevention or treatment benefit. This could be explored by postnatal genetic recombination.

The conditional deletion of PTP4A1 shown in Fig. 4 was indeed performed postnatally, in 7 week-old mice, just before the treatment with bleomycin. We apologize for the lack of clarity in the first version of the manuscript. This is now better explained in Fig. 4 and in the Material and Methods section.

5. Other TGFbeta regulated transcripts might have been expected to change in fibroblasts and the effect of TGFbeta stimulation should be explored. COL1A1 and COL2A1. Exploration at a protein level as these proteins not always transcriptionally regulated. CTGF is notably absent in the gene list described in the manuscript.

Following this reasonable concern, we have confirmed that silencing of PTP4A1 in NHDF leads to downregulation of other TGFbeta-regulated transcripts, including COL1A1 and CTGF. In addition, we measured collagen levels in the media obtained from the same NHDF lines, and found significantly reduced expression at the protein level (see **new Fig. 3C and S4**).

Overall, although this is very elegant work it is not clear how the findings are distinguished from those of many other studies that confirm activation and dysregulation of canonical and non-canonical TGFbeta signalling in SSc fibroblasts. This should be clarified in the abstract and conclusions of the paper.

We agree with the Reviewer that the concept that TGFbeta signaling is dysregulated in SSc fibroblasts is not novel. Indeed we believe that the novelty of our study consists in showing for

the first time that PTP4A1 is a TGFbeta-dependent gene and that it modulates TGFbeta signaling. We also propose a phosphatase activity-independent mechanism of action, which is absolutely novel in the phosphatase field. Considering that only very few phosphatases have been reported to play a role in fibrosis, and only one (PTPRA) is known to promote TGFbeta dependent fibrosis, we think our study is more significant for the advancement of the general field of TGFbeta-dependent fibrosis than specifically for SSc. We realized that the abstract and conclusions conveyed a potentially misleading message and have modified these sections accordingly.

Reviewer #2 (Remarks to the Author):

The present manuscript entitled “PTP4A1 phosphatase is overexpressed in systemic sclerosis fibroblasts and promotes TGFβ signaling” shows the implication of PTP4A1 (and PTP4A2) as molecular regulators of TGFβ signaling in healthy dermal fibroblasts and connects this, for PTP4A1, to the pathological context of systemic sclerosis.

Given the fact that protein tyrosine kinases are considered as potential therapeutic targets against SSc, this manuscript addresses the interesting question whether the molecular processes involved in SSc could also be modulated by protein tyrosine phosphatases. Among all PTPs, the authors have found PTP4A1 overexpressed in patient samples compared to healthy samples. PTP4A2 is also highly expressed, but not significantly elevated in patient samples. Therefore, the authors focus on PTP4A1 in the following study. These initial findings and the subsequent results showed in the manuscript are very relevant, adding novelty and contributing to a better understanding of the PTP4A group of phosphatases, which have been extensively described overexpressed in different types of cancer.

The novelties of this study include the following:

- 1) The involvement of PTP4A1 in TGFbeta signaling: This is robustly described, and is supported by previously published data such as promotion of SRC activity and MAPK signaling.
- 2) The differences regarding the involvement of PTP4A2 compared to PTP4A1 in this signaling process: These are largely clear.
- 3) The (potential) involvement in the disease. This is not so well established.
- 4) To me, the key impact of this work is in the direct interaction of PTP4A1 with SRC and its consequences, because many other indirect involvements in signaling pathways have been described previously (such as in MAPK and SRC signaling), leaving the question of how the phosphatase promotes signaling mechanistically. Because of the importance for the impact of the paper, the data should be completely convincing, which is currently not the case.

I therefore think the manuscript must be strengthened in order to be published in Nature Communications.

Major comments:

- 1) The authors mix up a healthy versus a disease background throughout the paper. They find PTP4A1 overexpressed in dcSScDF cells compared to NHDF cells. All molecular biology experiments on the mechanism are done in NHDF cells. They connect to the disease state again with their knock out mouse models by applying a bleomycin-induced dermal fibrosis test and note that the knock out mice are protected from skin fibrosis. However, no mechanistic

studies are done in the mouse. Then in the discussion (page 22, line 18,19) they state, "...we show that PTP4A1, a tyrosine phosphatase overexpressed in dermal fibroblasts of SSc patients, enhances canonical pro-fibrotic TGFbeta signaling in these cells." This is not correct as they show that it is involved in canonical pro-fibrotic TGFbeta signaling in NHDF cells from healthy humans. While it is very likely that PTP4A1 acts in SSc through this mechanism, it is not shown, and it could also act for example by taking over PTP4A2 actions or other processes dependent on the higher abundance of PTP4A1 in the cells. The authors need at least to check SRC levels (protein and phosphorylation on the two sites) in SSc cells in order to connect their studies in NHDF cells to the disease.

We agree with the points raised by the Reviewer and we have corrected the conclusion statements accordingly. In response to this comment, we also confirmed that SSc lines display overexpression of SRC protein and increased levels of activated ERK1/2, which is consistent with our working hypothesis (see **new Fig. S8C and S10A**).

2) In order to really connect PTP4A1-mediated stabilization of SRC to the Smad3 phenotype and MAPK signaling downstream, they should also use a SRC inhibitor in NHDF (and SSc) cells (for Fig. 7), like they did with Smad3 localization and ERK inhibition (Fig. S7 and S6). The inhibition of SRC should result in a similar phenotype and molecular profile as PTP4A1 knock down. This direct relation would strongly corroborate their working model.

We did the experiments requested by the Reviewer and found that incubation of NHDF lines with a SRC inhibitor led to inhibition of SMAD3 nuclear translocation profile similar to that observed following PTP4A1 knock down (see **new Fig. S10B**).

3) In figure 7, the influence of the catalytic activity of PTP4A1 in the TGFβ signaling is addressed. The luciferase assay including the catalytic mutant (7A) shows no effect of the PTP4A1 catalytic activity, when overexpressed, on the SMAD activity. To further demonstrate this, knockdown experiments are performed using an ASO (ASO3) targeting the exon where the crucial Arginine residue (Arg110) for the PTP4A1 catalysis is located. This is a rather unusual approach. Detailed information about how this approach works is missing in the text. My understanding of this approach (considering the information given in the supplementary figure S9) is that the ASO3 generates a truncated protein that lacks the exon 4 sequence including the Arg110, and the resulting protein is then presumably inactive. If this were correct, I would have the concern of the stability of the protein generated upon ASO3 treatment, since creating an internal deletion could lead to a misfolded and less stable protein. If that were the case, the results shown in Figure 7B, 7C and 7D where the knockdown with ASO3 does not give any effect and behaves similar to the control ASO, could then be also explained by a putative unstable/misfolded/still catalytically active protein mimicking wt PTP4A1 behavior rather than by a truncated protein only lacking catalytic activity. In order to confirm this, the presence of the truncated protein has to be shown (by MS or Western blot), the stability and folding (for example by CD and Tm measurement) of the resulting truncated protein should be studied and compared to the wt PTP4A1. It should also be made sure that the truncated protein does not have

phosphatase activity, and that it still binds to SRC, as this is the proposed mechanism. Alternatively, an inhibitor for the PTP4A family could be used. Since the authors can distinguish PTP4A1 and 2 by the Smad3 read out, they can use a promiscuous PTP4A inhibitor like thienopyridone to test for inhibited PTP4A1 action.

We understand the Reviewer's concern and performed additional experimentation, which we hope will increase her/his level of confidence in the data produced with the truncation mutant of PTP4A1. We isolated the truncated PTP4A1 as a recombinant protein and confirmed that in vitro it does not have phosphatase activity, compared to wt PTP4A1 (see **new Fig. S11B**). We also confirmed that the mutant retains its ability to co-precipitate with SRC (see **new Fig. S12B**). Following the Reviewer's recommendation, we also looked for small molecule inhibitors of PTP4A1 activity. We were unable to obtain thienopyridone, but we tested the effect of Cmpd43, an inhibitor of PTP4A1 trimerization, which is supposed to indirectly inhibit the activity of PTP4A1 (Bai Y, Cancer Res 2016; 76:4805-15). We saw no effect on TGFbeta signaling in NHDF. We will be happy to make the data available to the Reviewers upon request.

4) Figure 6E shows that the knockdown of PTP4A1 increases the phosphorylation levels of SRC at Tyr416 and Tyr527, indicating a regulation of SRC activation mediated by PTP4A1. The authors further investigate whether Tyr416 is indeed dephosphorylated by PTP4A1, detecting no dephosphorylation (fig. 7E). However, they do not show a similar experiment to see if SRC Tyr527 is a substrate of PTP4A1 in NHDF lines. Since the dephosphorylation of Tyr527 leads to the activation of SRC, it is worth to see if this residue is a substrate of PTP4A1 because this would be also one explanation of how SRC is kept active by PTP4A1. The authors detect the physical interaction between SRC and PTP4A1 and suggest the hypothesis of this interaction as a regulatory mechanism of SRC activation (preventing SRC basal autophosphorylation that would lead to an increase in SRC protein degradation), but it should be taken into account that a putative dephosphorylation of SRC Tyr527 by PTP4A1 could influence the state of SRC activation as well. This is necessary to really "clarify the relationship between SRC and PTP4A1" (page 23 line 4).

This is also a very good point. In response to this comment, we assessed whether PTP4A1 can dephosphorylate SRC after in vitro phosphorylation with CSK and we found that PTP4A1 is unable to dephosphorylate SRC-pTyr527 (see **new Fig. S14B**).

5) Figure S10: It is standard when showing a new interaction to do a reciprocal IP. The interaction has only been shown through Src IP, but the HA-PTP4A1 should be IPed as well. If the interaction is not affected by phosphatase activity (as they imply through using the C/S mutant in cells), why do they use the C/S mutants in the IP? Since PTP4A2 also binds to Src, do the authors think that there is an overlap here with their mechanisms after all? Or is it background? The respective reciprocal IP could also clarify this.

In response to this concern, we confirmed the specific interaction between wtPTP4A1 and SRC with a trapping assay, and performed the requested reciprocal IP of HA-tagged PTP4A1/A2 in 293T cells and confirmed that HA-tagged wtPTP4A1 co-precipitates with SRC much more than wtPTP4A2 (see **new Fig. S12A and C**).

Minor comments:

1) Figures 5, 6 and 7 show in some cases the quantification of the phosphorylation levels (normalized to the corresponding total protein amount) of different proteins. However, the label in the y-axes refer to Relative expression and this term is not accurate in this case. I suggest to change the axis labels to “relative phosphorylation levels”, for example.

We agree and changed “Relative expression” to “Relative phosphorylation levels” in all the graphs that show quantification of phosphorylation levels.

2) Unlike PTP4A1, PTP4A2 does not regulate the expression levels of SMAD3 (fig. 3A), does not affect the SMAD3 nuclear translocation (fig. S7B) and does not promote ERK1/2 activation (fig. 5D). As pointed out by the authors, these observations indicate different pathways between PTP4A1 and PTP4A2. Nevertheless, although small, PTP4A2 enhances (1.6 fold) the luciferase activity (fig. 5E). I would like to know the authors opinion about how PTP4A2 could enhance the activity of SMAD3 (see also the detected interaction between PTP4A2 and SRC, mentioned above).

This is a good question. We speculate that PTP4A2 either does interact a little bit with SRC -but much less than PTP4A1- in the 293 overexpression assay, or it might indirectly activate the SMAD3 pathway via promotion of the ROCK pathway (see Fig. 5F).

3) Discussion, page 18, line 17: typo: correct TFG by TGF

We have corrected the typo.

4) Discussion, page 22, line 12: “Src-inhibitory activity of PTP4A1” is a bit confusing, since in cells the overall outcome is more presence and activity of Src. It would be good to rephrase this somehow.

We agree, we have rephrased and in the revised text we avoid using the term “inhibitory”.

5) Two previous studies showed that PTP4A3 (the third member of the PTP4A phosphatases, not expressed in SScDF) exerts translational control of CSK, resulting in SRC activation (Liang et al., J Biol Chem. 2007 Feb 23;282(8):5413-9 and J Biol Chem. 2008 Apr 18;283(16):10339-46). Could PTP4A1 regulate CSK in the same way turning into SRC activation? This fact should be mentioned in the discussion part about SRC regulation. It would be interesting to reflect that these two PTP4A phosphatases could potentially have different or similar ways of regulate SRC (see also potential dephosphorylation of Y527).

This is an excellent point and we have expanded the discussion to include these considerations. We also have analyzed CSK mRNA and protein levels in and we found no differences after PTP4A1-silencing (see **new Fig. S14A**).

6) Introduction: page 4 line 10: There is evidence that within the PTP4A family there are differences between them, such as promotion of EMT by PTP4A3, their different expression profiles (related to cell-type dependent actions as also mentioned in this manuscript), and their reported different in vitro substrate specificities. This sentence should be changed, as it is too general.

The sentence has been rephrased.

7) Page 7 lines 1-12: What is the difference between the early and late biopsies? What could be the reason that PTP4A1 overexpression is not evident at later stages?

The inflammatory component of SSc pathogenesis is more evident in the early stages of disease. Since we found that PTP4A1 expression is enhanced by TGFbeta (see Fig. S2B), we speculate that enhanced production of TGFbeta drives more expression of PTP4A1 in early vs late stages. However, it is also possible that immune-derived cytokines contribute to the observed difference between early and late disease. These considerations are now mentioned in the revised discussion.

8) For all knock-downs, the Western blots should be shown.

Western blots are now shown that display the depth of knockdown of PTP4A1 and PTP4A2 achieved with their respective ASOs (see **new Fig. S5B**).

In conclusion, this work has the novelty, insight and impact to be considered for publication in Nature Communications, but these concerns should be addressed thoroughly beforehand.

We thank the Reviewers, and hope that they will now find our manuscript substantially improved!

REVIEWERS' COMMENTS:

Reviewer #1 (Remarks to the Author):

The manuscript is substantially revised and the points that I raised in my review have been addressed or clarified.

Reviewer #2 (Remarks to the Author):

The authors have thoroughly addressed my comments, and in my opinion the work can be published in this form. I have no further concerns.